# Information Theoretic Guarantees For Policy Alignment In Large Language Models

**Youssef Mroueh**                                                    *mroueh@us.ibm.com*
*IBM Research*

**Apoorva Nitsure**                                               *Apoorva.Nitsure@ibm.com*
*IBM Research*

**Reviewed on OpenReview:** *https: // openreview. net/ forum? id= Uz9J77Riul*

## Abstract

Policy alignment of large language models refers to constrained policy optimization, where the policy is optimized to maximize a reward while staying close to a reference policy based on an $f$-divergence like $\mathsf{KL}$ divergence. The best of $n$ alignment policy selects the sample with the highest reward from $n$ independent samples. Recent work shows that the reward improvement of the aligned policy scales as $\sqrt{\mathsf{KL}}$, with an explicit bound on the $\mathsf{KL}$ for best of $n$ policies. We show that this $\sqrt{\mathsf{KL}}$ bound holds if the reference policy's reward has sub-gaussian tails. For best of $n$ policies, the $\mathsf{KL}$ bound applies to any $f$-divergence through a reduction to exponential order statistics using the Rényi representation. Tighter control can be achieved with Rényi divergence if additional tail information is known. Finally, we demonstrate how these bounds transfer to golden rewards, resulting in decreased golden reward improvement due to proxy reward overestimation and approximation errors.

## 1 Introduction

Aligning Large Language Models (LLMs) with human preferences allows a tradeoff between maintaining the utility of the pre-trained reference model and the alignment of the model with human values such as safety or other socio-technical considerations. Alignment is becoming a crucial step in LLMs training pipeline, especially as these models are leveraged in decision making as well as becoming more and more accessible to the general public. Policy alignment starts by learning a reward model that predicts human preferences, these reward models are typically fine-tuned LLMs that are trained on pairwise human preference data (Christiano et al., 2017; Stiennon et al., 2020; Ouyang et al., 2022; Bai et al., 2022). The reward is then optimized using *training time alignment* i.e via policy gradient based reinforcement learning leading to the so called *Reinforcemnent Learning from Human Feedback* (RLHF) (Christiano et al., 2017). RLHF ensures that the reward is maximized while the policy $\pi$ stays close to the initial reference policy $\pi_{\mathrm{ref}}$ in the sense of the Kullback-Leibler divergence $\mathsf{KL}(\pi||\pi_{\mathrm{ref}})$. Other variants of these training time alignment have been proposed via direct preference optimization (Rafailov et al., 2024) (Zhao et al., 2023) (Ethayarajh et al., 2024). Another important paradigm for optimizing the reward is *test time alignment* via best of $n$ sampling from the reference policy and retaining the sample that maximizes the reward. The resulting policy is known as the *best of $n$ policy*. The best of $n$ policy is also used in controlled decoding settings (Yang & Klein, 2021; Mudgal et al., 2023) and in fine-tuning LLMs to match the best of $n$ policy responses (Touvron et al., 2023).

Gao et al. (2023) and Hilton & Gao (2022) studied the scaling laws of reward models optimization in both the RL and the best of $n$ setups. Gao et al. (2023) distinguished between "golden reward" that can be thought of as the golden human preference and "proxy reward" which is trained to predict the golden reward. For proxy rewards Gao et al. (2023) found experimentally for both RL and best of $n$ policies that the reward improvement on the reference policy scales as $\sqrt{\mathsf{KL}(\pi||\pi_{\mathrm{ref}})}$. Similar observations for reward improvement scaling in RL were made in (Bai et al., 2022). For golden rewards, Gao et al. (2023) showed for both RL and best of $n$

policies that LLMs that optimize the proxy reward suffer from over-optimization in the sense that as the policy drifts from the reference policy, optimizing the proxy reward results in deterioration of the golden reward.

This phenomena is referred to in (Gao et al., 2023) (Hilton & Gao, 2022) as Goodhart's law. A qualitative plot of scaling laws discovered in (Gao et al., 2023) is given in Figure 1. For the best of $n$ policy, most works in this space assumed that $\mathsf{KL}(\pi||\pi_{\mathrm{ref}}) = \log(n) - \frac{n-1}{n}$ (Stiennon et al., 2020; Coste et al., 2024; Nakano et al., 2021; Go et al., 2024; Gao et al., 2023). Recently Beirami et al. (2024) showed that this is in fact an inequality under the assumption that the reward is one to one map (a bijection) and for finite alphabets.

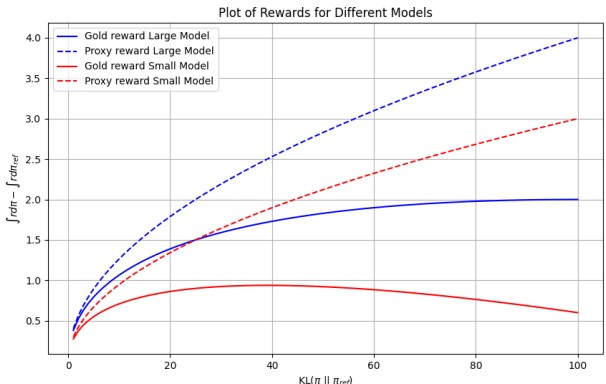

Figure 1: Qualitiative plot of centered rewards vs. KL of Proxy and Gold Rewards for both Best of $n$ and RL policies. (See Fig. 1 a) and b) in (Gao et al., 2023) for scaling laws in policy alignment).

The main contributions of this paper are :

1. In Theorem 1 (Section 2), we provide a new proof for the best of $n$ policy inequality $\mathsf{KL}(\pi||\pi_{\mathrm{ref}}) \leq \log(n) - \frac{n-1}{n}$, showing it results from the data processing inequality of $\mathsf{KL}$. We extend this beyond Beirami et al. (2024)'s setup of one-to-one rewards and finite alphabets to surjective rewards and beyond finite alphabets, also giving conditions for equality and generalizing to $f$-divergences and Rényi divergences.

2. In Section 3, we show that the policy improvement scaling laws in (Gao et al., 2023) are information-theoretic upper bounds, derived from transportation inequalities with $\mathsf{KL}$ under sub-gaussian reward tails. We discuss how the $\mathsf{KL}$ dependence is driven solely by the reward tails and can only improve if they are fatter than sub-gaussian (e.g., sub-gamma or sub-exponential).

3. In Theorem 4, we examine the tightness of these bounds when the optimized policy's tails are known, deriving new transportation inequalities for Rényi divergence $D_\alpha$ for $\alpha \in (0,1)$. We show that the $\sqrt{\mathsf{KL}}$ upper bound cannot be met, reflecting Goodhart's law (Gao et al., 2023).

4. We show in Section 3.3, that $\sqrt{\mathsf{KL}}$ policy improvement upper bounds for best of $n$ polices with bounded and calibrated rewards are loose and tighter upper bounds can be obtained leveraging our best of $n$ upper bounds for the total variation distance.

5. In Section 4, we study the transfer of transportation inequalities from proxy to golden rewards, proving that golden reward improvement is limited by the proxy reward's overestimation, as reported empirically in (Gao et al., 2023).

Our work answers positively the following question:

*"Without performing alignment, can we predict an upper bound on the expected reward for a given $\mathsf{KL}$ level given the distribution of the reward of the reference model?" Indeed from the tails of the reward under the reference model we can predict the upper bound for alignment as shown in Figures 7 and 8.*

## 2 The Alignment Problem

### 2.1 RLHF: A Constrained Policy Optimization Problem

Let $\mathcal{X}$ be the set of prompts and $\mathcal{Y}$ be the set of responses $y \in \mathcal{Y}$ from a LLM conditioned on a prompt $x \in \mathcal{X}$. The reference LLM is represented as policy $\pi_{\text{ref}}(y|x)$, i.e as a conditional probability on $\mathcal{Y}$ given a prompt $x \in \mathcal{X}$. Let $\rho_{\mathcal{X}}$ be a distribution on prompts, and a $r$ a reward, $r : \mathcal{X} \times \mathcal{Y} \to \mathbb{R}$, $r$ represents a safety or alignment objective that is desirable to maximize.

Given a reference policy $\pi_{\text{ref}}$, the goal of alignment is to find a policy $\pi^*$ that maximizes the reward $r$ and that it is still close to the original reference policy for some positive $\Delta > 0$:

$$\pi^*_{y|x} = \arg \max_{\pi_{y|x}} \mathbb{E}_{x \sim \rho_{\mathcal{X}}} \mathbb{E}_{y \sim \pi(.|x)} r(x, y)$$

$$\text{s.t} \int_{\mathcal{X}} \mathsf{KL}(\pi(y|x)||\pi_{\text{ref}}(y|x)) d\rho_{\mathcal{X}}(x) \leq \Delta, \tag{1}$$

where $\mathsf{KL}(\pi(y|x)||\pi_{\text{ref}}(y|x)) = \mathbb{E}_{y \sim \pi(\cdot|x)} \log \left( \frac{\pi(y|x)}{\pi_{\text{ref}}(y|x)} \right)$. With some abuse of notation, we write $\pi(x, y) = \pi(y|x)\rho_{\mathcal{X}}(x)$ and $\pi_{\text{ref}}(x, y) = \pi_{\text{ref}}(y|x)\rho_{\mathcal{X}}(x)$. Let $\mathcal{P}(\mathcal{X} \times \mathcal{Y})$ be joint probability defined on $\mathcal{X} \times \mathcal{Y}$ that has $\rho_{\mathcal{X}}$ as marginal on $\mathcal{X}$. Hence we can write the alignment problem equation 1 in a more compact way as follows:

$$\sup_{\pi \in \mathcal{P}(\mathcal{X} \times \mathcal{Y})} \int r d\pi \quad \text{s.t } \mathsf{KL}(\pi||\pi_{\text{ref}}) \leq \Delta. \tag{2}$$

For $\gamma > 0$, we can also write a penalized form of this constrained policy optimization problem as follows[1]:

$$\sup_{\pi \in \mathcal{P}(\mathcal{X} \times \mathcal{Y})} \int r d\pi - \frac{1}{\gamma} \mathsf{KL}(\pi||\pi_{\text{ref}}).$$

It is easy to see that the optimal policy of the penalized problem is given by:

$$\pi_{\gamma, r}(y|x) = \frac{\exp(\gamma r(x, y))\pi_{\text{ref}}(y|x)}{\int \exp(\gamma r(x, y)) d\pi_{\text{ref}}(y|x)} \quad \rho_{\mathcal{X}} \text{almost surely.} \tag{3}$$

The constrained problem equation 2 has a similar solution (See for e.g (Yang et al., 2024)):

$$\pi_{\lambda_{\Delta}, r}(y|x) = \frac{\exp(\frac{r(x,y)}{\lambda_{\Delta}})\pi_{\text{ref}}(y|x)}{\int \exp(\frac{r(x,y)}{\lambda_{\Delta}}) d\pi_{\text{ref}}(y|x)} \quad \rho_{\mathcal{X}} \text{almost surely,} \tag{4}$$

where $\lambda_{\Delta} > 0$ is a lagrangian that satisfies $\int_{\mathcal{X}} \mathsf{KL}(\pi_{\lambda_{\Delta}, r}(y|x)||\pi_{\text{ref}}(y|x)) d\rho_{\mathcal{X}}(x) = \Delta$.

### 2.2 Best of $n$ Policy Alignment

Let $X$ be the random variable associated with prompts such that $\text{Law}(X) = \rho_{\mathcal{X}}$. Let $Y$ be the random variable associated with the conditional response of $\pi_{\text{ref}}$ given $X$. Define the conditional reward of the reference policy :

$$R(Y)|X := r(X, Y) \text{ where } Y \sim \pi_{\text{ref}}(.|X),$$

we assume that $R(Y)|X$ admits a CDF denoted as $F_{R(Y)|X}$ and let $F^{-1}_{R(Y)|X}$ be its quantile:

$$F^{(-1)}_{R(Y)|X}(p) = \inf\{\eta : F_{R(Y)|X}(\eta) \geq p\} \text{ for } p \in [0, 1].$$

Let $Y_1 \dots Y_n$ be independent samples from $\pi_{\text{ref}}(.|X)$. We define the best of $n$ reward as follows:

$$R^{(n)}(Y)|X = \max_{i=1\dots n} R(Y_i)|X, \tag{5}$$

this the maximum of $n$ iid random variables with a common CDF $F_{R(Y)|X}$. The best of $n$ policy corresponds to $Y^{(n)}|X := \arg\max_{i=1\dots n} r(X, Y_i)$. We note $\pi^{(n)}_{r,\text{ref}}(.|X)$ the law of $Y^{(n)}|X$. $\pi^{(n)}_{r,\text{ref}}$ is referred to as the best of $n$ alignment policy. We consider two setups for the reward:

---

[1]The regularizer $\frac{1}{\gamma}$ is usually referred to as $\beta$.

**Assumption 1** *We assume that the reward $r$ is a one to one map for a fixed $x$, and admits an inverse $h_x : \mathbb{R} \to \mathcal{Y}$ such that $h_x(r(x,y)) = y$.*

This assumption was considered in (Beirami et al., 2024). Nevertheless this assumption is strong and not usually met in practice, we weaken this assumption to the following:

**Assumption 2** *We assume that there is a stochastic map $H_X$ such that $H_X(R_{Y|X}) \overset{d}{=} Y|X$ and $H_X(R_{Y^{(n)}|X}) \overset{d}{=} Y^{(n)}|X$.*

Under Assumption 2, the reward can be surjective which is more realistic but we assume that there is a stochastic map that ensures invertibility not point-wise but on a distribution level. Our assumption means that we have conditionally on $X$: $R|X \to Y|X$ form a markov chain i.e exists $A(Y|R,X)$ so that $P_{Y|X} = A(Y|R,X)P_{R|X}$, and $P_{Y^{(n)}|X} = A(Y|R,X)P_{R^{(n)}|X}$. *Note that Assumption 2, will always hold from Bayes rule as long as this Markov kernel $A$ is well defined.*

**Best of $n$ Policy KL Guarantees: A reduction to Exponentials and Data Processing Inequalities** In what follows for random variables $Z, Z'$ with laws $p_Z, p_{Z'}$ we write interchangeably: $\mathsf{KL}(p_Z||p_{Z'}) = \mathsf{KL}(Z||Z')$.

Our goal in this section is to relate the Kullback-Leibler (KL) divergence $\mathsf{KL}(Y^{(n)}||Y|X)$, which measures the difference between the best of $n$ policy $(Y^{(n)}|X)$ and the reference policy $(Y|X)$, to $\mathsf{KL}(R^{(n)}(Y)||R(Y)|X)$, which quantifies the divergence between the reward distribution under the best of $n$ policy $(R^{(n)}(Y)|X)$ and the reward distribution under the reference policy $(R(Y)|X)$. Another objective is to express this latter divergence as a function of $n$. To achieve this, we will leverage a powerful tool in information theory known as Strong Data Processing Inequalities (DPI).

In information theory, the Data Processing Inequality (DPI) (Polyanskiy & Wu, 2023) establishes a fundamental limit on how information evolves under transformations. It states that the Kullback-Leibler (KL) divergence between two random variables, $U$ and $U'$, cannot increase after applying the same transformation to both. This transformation can be either a deterministic function $h$ or a probabilistic mapping, commonly referred to as a "channel" and denoted by $H = P_{V|U}$. Formally, DPI asserts that:

$$\mathsf{KL}(V||V') \leq \mathsf{KL}(U||U'), \tag{6}$$

where $V$ and $V'$ are the transformed versions of $U$ and $U'$, respectively. This ensures that information content, as measured by KL divergence, cannot increase through processing. DPI is tight if $h$ is a one to one function (Polyanskiy & Wu, 2023).

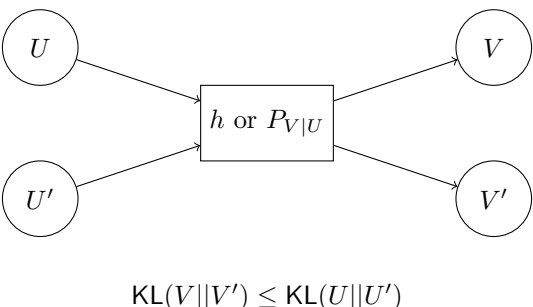

$$\mathsf{KL}(V||V') \leq \mathsf{KL}(U||U')$$

Figure 2: Data Processing Inequality for the KL divergence.

Under Assumption 1, the reward function is a bijection, allowing us to transform the rewards $R^{(n)}(Y)|X = R(Y^{(n)})|X$ and $R(Y)|X$ into $Y^{(n)}|X$ and $Y|X$ via the inverse mapping $h_X$ of $r(X,.)$. On the other hand, under Assumption 2, this transformation is achieved through the Markov kernel $A(Y|R,X)$.

These transformations are illustrated in Figure 3. By applying the strong Data Processing Inequality for the $\mathsf{KL}$ divergence, we obtain:

$$\mathsf{KL}\left(\pi_{r,\mathrm{ref}}^{(n)}||\pi_{\mathrm{ref}}|X\right) = \mathsf{KL}(Y^{(n)}||Y|X) \leq \mathsf{KL}(R^{(n)}(Y)||R(Y)|X). \tag{7}$$

Inequality equation 7 is tight under Assumption 1.

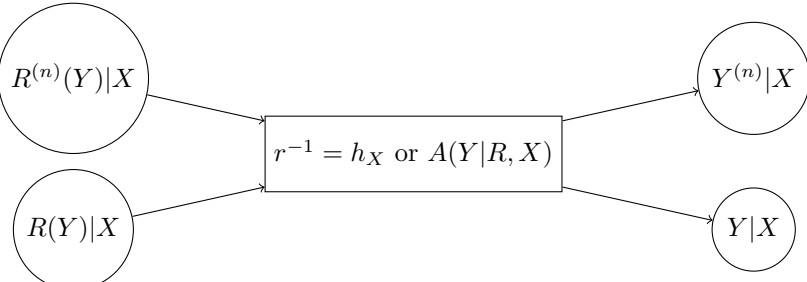

Figure 3: From rewards to outputs via $r^{-1}$ under Assumption 1 or via the stochastic map (Markov kernel $A$) under Assumption 2.

Turning to $\mathsf{KL}(R^{(n)}(Y)||R(Y)|X)$, our goal is to bound or express it as a function of $n$. To achieve this, we aim to leverage the Data Processing Inequality (DPI). Specifically, we seek to transform a pair of random variables, for which the $\mathsf{KL}$ divergence has a known analytical expression in terms of $n$, into $R^{(n)}(Y)|X$ and $R(Y)|X$, respectively.

Transporting one random variable to another is at the core of optimal transport theory. In the following, we demonstrate that an exponential random variable $E$ and the maximum of $n$ i.i.d. exponentials, $E^{(n)}$, allow such transformation via the optimal transport map.

Let $E \sim \mathrm{Exp}(1)$. The optimal transport map $T_X = F_{R(Y)|X}^{-1} \circ F_E$ from the exponential distribution $E$ to $R(Y)|X$ (see, for example, Theorem 2.5 in (Santambrogio, 2015), noting that $E$ is atomless while $R(Y)|X$ can take discrete values) allows us to express:

$$R(Y)|X \overset{d}{=} T_X(E), \tag{8}$$

where $\overset{d}{=}$ denotes equality in distribution. This representation illustrates that the target reward distribution can be generated through the optimal transport map $T_X = F_{R(Y)|X}^{-1} \circ F_E$ as illustrated below:

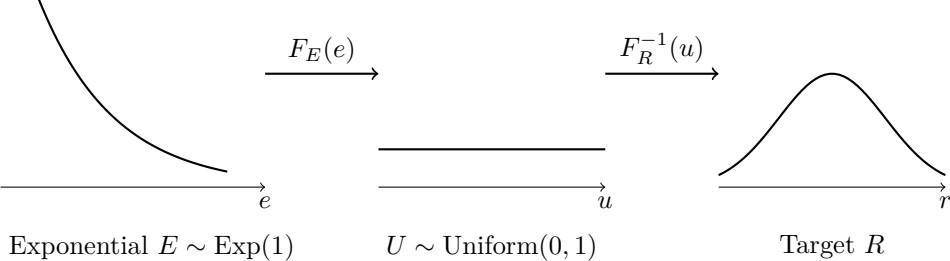

Figure 4: Optimal tansport map from an Exponentially distributed random variable $E$ to the random variable $R$ representing the reward.

On the other hand, let $R^{(1)}(Y)|X \leq \cdots \leq R^{(n)}(Y)|X$ be the order statistics of the rewards of $n$ independent samples $Y_i, i = 1 \ldots n$, $Y_i \sim \pi_{\mathrm{ref}}(.|X)$. The order statistics refer to sorting the random variable from the minimum (index (1)) to the maximum (index $(n)$). Consider $n$ independent exponential $E_1, \ldots E_n$, where $E_i \sim Exp(1)$, and their order statistics $E^{(1)} \leq E^{(2)} \leq \ldots E^{(n)}$. The Rényi representation of order statistics

(Rényi, 1953), similar to the Optimal Transport (OT) representation allows us to express the distribution of the order statistics of the rewards in terms of the order statistics of exponentials as follows:

$$\left(R^{(1)}(Y)|X,\ldots,R^{(n)}(Y)|X\right) \stackrel{d}{=} \left(F_{R(Y)|X}^{-1} \circ F_E(E^{(1)}),\ldots,F_{R(Y)|X}^{-1} \circ F_E(E^{(n)})\right)$$
$$\stackrel{d}{=} \left(T_X(E^{(1)}),\ldots T_X(E^{(n)})\right) \tag{9}$$

The central idea in the Rényi representation is that the mapping $T_X = F_{R(Y)|X}^{-1} \circ F_E$ is monotonic and hence ordering preserving and by the OT representation each component is distributed as $R(Y)|X$. See (Boucheron & Thomas, 2012) for more account on the Rényi representation of order statistics. Note that we could have used uniform random variables instead of exponential, we use exponentials to stay faithful to Rényi representation as exponential order statistics have nice properties.

Thanks to the mapping $T_X$, we can transform an exponentially distributed random variable $E$ as well as the maximum of $n$ iid Exponentials $E^{(n)}$ to $R(Y)|X$ and $R^{(n)}(Y)|X$ respectively. This transformation is illustrated in Figure 5.

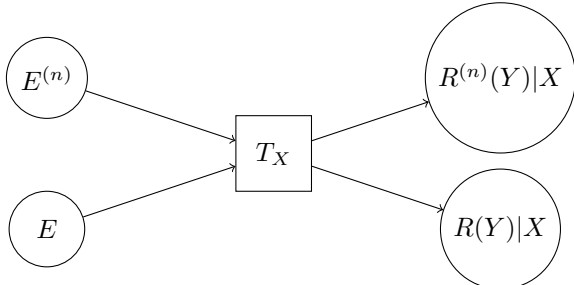

Figure 5: Transforming exponential and maximum of exponentials to the reward and maximum reward random variables via the optimal transport map $T_X$, by virtue of the OT representation equation 8, and the Rényi representation equation 9 respectively. We recognize $T_X$ as the "channel" in Data Processing Inequalities.

Hence using the OT representation in equation 8 and the Rényi representation of the maximum equation 9 and the data processing inequality, we can bound the KL between the rewards to a KL on functions of exponentials and their order statistics:

$$\mathsf{KL}\left(R^{(n)}(Y)||R(Y)\Big|X\right) = \mathsf{KL}(T_X(E^{(n)})||T_X(E)) \leq \mathsf{KL}(E^{(n)}||E), \tag{10}$$

the tightness of this inequality depends on the properties of the map $T_X$ and the space $\mathcal{Y}$.

Hence chaining the two data processing inequalities given in equation 7 and equation 10 we obtain finally :

$$\mathsf{KL}\left(\pi_{r,\mathrm{ref}}^{(n)}||\pi_{\mathrm{ref}}|X\right) \leq \mathsf{KL}\left(R^{(n)}(Y)||R(Y)\Big|X\right) \leq \mathsf{KL}(E^{(n)}||E), \tag{11}$$

tightness of each inequality depends on the tightness of each DPI.

The following Lemma gives a closed form expression for $\mathsf{KL}(E^{(n)}||E)$:

**Lemma 1 (KL Between Exponential and Maximum of Exponentials)** *Let* $E \sim Exp(1)$, *and* $E_1,\ldots E_n$ *be iid exponentials and* $E^{(n)}$ *their maximum, we have:*

$$\mathsf{KL}(E^{(n)}||E) = \log(n) - \frac{n-1}{n}. \tag{12}$$

The following result establishes the the best of $n$ policy KL expression or bounds in terms of $n$ :

**Theorem 1** *The best of n policy satisfies under (i) Assumption 1 (reward one to one) and for finite $\mathcal{Y}$ or under (ii) Assumption 2 (existence of stochastic "inverse") :*

$$\mathsf{KL}(\pi_{r,\text{ref}}^{(n)}||\pi_{\text{ref}}) \leq \mathsf{KL}(E^{(n)}||E) = \log(n) - \frac{n-1}{n}. \tag{13}$$

*Under Assumption 1, for infinite $\mathcal{Y}$ and assuming $F_{R(Y|X)}$ is continuous and strictly increasing for all $X$ we have:*

$$\mathsf{KL}(\pi_{r,\text{ref}}^{(n)}||\pi_{\text{ref}}) = \mathsf{KL}(E^{(n)}||E) = \log(n) - \frac{n-1}{n}. \tag{14}$$

| Divergence | $f(x)$ | Bound on $\mathrm{D}_f(\pi_{r,\text{ref}}^{(n)}||\pi_{\text{ref}})$ |
|---|---|---|
| KL | $x\log(x)$ | $\log(n) - \frac{n-1}{n}$ |
| Chi-squared | $(x-1)^2$ | $\frac{(n-1)^2}{2n-1}$ |
| Total Variation | $f(x) = \frac{1}{2}|x-1|$ | $\left(\frac{1}{n}\right)^{\frac{1}{n-1}} - \left(\frac{1}{n}\right)^{\frac{n}{n-1}}$ |
| Hellinger distance | $(1-\sqrt{x})^2$ | $2\frac{(1-\sqrt{n})^2}{n+1}$ |
| Forward KL | $-\log(x)$ | $n - 1 - \log(n)$ |
| $\alpha$ Rényi Divergence | NA | $\frac{1}{(\alpha-1)}\log\left(\frac{n^\alpha}{\alpha(n-1)+1}\right)$ |

Table 1: Best of $n$ policy $f$-Divergence and $\alpha$ Rényi Divergence Bounds.

Beirami et al. (2024) showed this result under condition (i) which is not a realistic setting and used the finiteness of $\mathcal{Y}$ to provide a direct proof. Our analysis via chaining DPI and using OT and Rényi representations to reduce the problem to exponentials allows us to extend the result to a more realistic setup under condition (ii) i.e the existence of a stochastic "inverse", without any assumption on $\mathcal{Y}$. Furthermore our analysis sheds the light on the underlying assumptions of invertible reward and continuous space $\mathcal{Y}$ that was implicitly made in previous works that stated this bound as an equality (Stiennon et al., 2020) (Coste et al., 2024; Nakano et al., 2021; Go et al., 2024) (Hilton & Gao, 2022) (Gao et al., 2023). Singh et al. (2025) recently built on this result to analyze the KL divergence of best of $n$ in diffusion models.

Our approach of reduction to exponentials using Rényi representation of order statistics and data processing inequalities extends to bounding the $f$-divergence $\mathrm{D}_f(\pi_{r,\text{ref}}^{(n)}||\pi_{\text{ref}})$ as well as the $\alpha$ Rényi divergence. The Rényi divergence for $\alpha \in (0,1) \cup (1,\infty)$ is defined as follows:

$$D_\alpha(P||Q) = \frac{1}{(\alpha-1)}\log\left(\int p^\alpha(x)q^{1-\alpha}(x)dx\right)$$

the limit as $\alpha \to 1$ coincides with KL, i.e: $D_1(P||Q)) = \mathsf{KL}(P||Q)$. These bounds are summarized in Table 1. Full proofs and theorems are in the Appendix.

**Best of $n$-Policy Dominance on the Reference Policy.** The following proposition shows that the best of $n$ policy leads to an improved reward on average:

**Proposition 1** *$R^{(n)}$ dominates $R$ in the first order dominance that is $R^{(n)}$ dominates $R$ on all quantiles: $Q_{R^{(n)}}(t) \geq Q_R(t), \forall t \in [0,1]$. It follows that we have $\mathbb{E}R^{(n)} \geq \mathbb{E}R$.*

**Best of $n$ Policy and RL Policy** The following proposition discusses the sub-optimality of the best of $n$ policy with respect to the alignment RL objective given in equation 1:

**Proposition 2** *Assume a bounded reward in $[-M, M]$. For $\Delta > 0$ and $n = \exp(\Delta)$ the best of n policy $\pi_{r,\text{ref}}^{(n)}$ and the $\Delta$ Constrained RL policy $\pi_{\lambda_\Delta,r}$ (given in equation 4) satisfy:*

$$\mathsf{KL}(\pi_{r,\text{ref}}^{(n)}||\pi_{\lambda_\Delta,r}) \leq \frac{\sqrt{2\pi}M(e^{\frac{2M}{\lambda_\Delta}}-1)}{\lambda_\Delta}\exp\left(-\frac{\Delta}{2}\right).$$

A similar asymptotic result appeared in (Yang et al., 2024) for $\Delta \to \infty$, showing as $n \to \infty$, $\mathsf{KL}(\pi_{r,\mathrm{ref}}^{(n)}||\pi_{\lambda_\Delta,r}) \to 0$, we provide here a non asymptotic result for finite $n$ and finite $\Delta$. This result shows that the best of $n$ policy can be seen as approximation of the RL policy for $n = \exp(\Delta)$, where $\Delta$ is the desired $\mathsf{KL}$ level.

## 3 Reward Improvement Guarantees Through Transportation Inequalities

**Notations** Let $X$ be a real random variable. The logarithmic moment generating function of $X$ is defined as follows for $\lambda \in \mathbb{R}$: $\psi_X(\lambda) = \log \mathbb{E}_X e^{\lambda(X - \mathbb{E}X)}$. $X$ is said to be sub-Gaussian with variance $\sigma^2$ if : $\psi_X(\lambda) \leq \frac{\lambda^2 \sigma^2}{2}$ for all $\lambda \in \mathbb{R}$. We denote $\mathsf{SubGauss}(\sigma^2)$ the set of sub-Gaussian random variables with variance $\sigma_{\mathrm{ref}}^2$. $X$ is said to be sub-Gamma on the right tail with variance factor $\sigma^2$ and a scale parameter $c > 0$ if : $\psi_X(\lambda) \leq \frac{\lambda^2 \sigma^2}{2(1-c\lambda)}$ for every $\lambda$ such that $0 < \lambda < \frac{1}{c}$. We denote $\mathsf{SubGamma}(\sigma^2, c)$ the set of left and right tailed sub-Gamma random variables. Sub-gamma tails can be thought as an interpolation between sub-Gaussian and sub-exponential tails.

**Scaling Laws in Alignment** It has been observed empirically (Coste et al., 2024; Nakano et al., 2021; Go et al., 2024; Hilton & Gao, 2022; Gao et al., 2023) that optimal RL policy $\pi_{\lambda_\Delta,r}$ satisfy the following inequality for a constant $\sigma_{\mathrm{ref}}^2$ estimated from data :

$$\mathbb{E}_{\pi_{\lambda_\Delta,r}} r - \mathbb{E}_{\pi_{\mathrm{ref}}} r \leq \sqrt{2\sigma_{\mathrm{ref}}^2 \mathsf{KL}(\pi_{\lambda_\Delta,r}||\pi_{\mathrm{ref}})}.$$

A similar scaling for best of $n$ policy :

$$\mathbb{E}_{\pi_{r,\mathrm{ref}}^{(n)}} r - \mathbb{E}_{\pi_{\mathrm{ref}}} r \leq \sqrt{2\sigma_{\mathrm{ref}}^2 \left( \log n - \frac{n-1}{n} \right)},$$

and those bounds are oftentimes tight even when empirically estimated from samples.

**The case of Bounded Rewards and Pinsker Inequality** This hints that those bounds are information theoretic and independent of the alignment problem. Indeed if the reward was bounded, a simple application of Pinsker inequality gives rise to $\sqrt{\mathsf{KL}}$ scaling. Let $\mathsf{TV}$ be the total variation distance, we have: $\mathsf{TV}(\pi, \pi_{\mathrm{ref}}) = \frac{1}{2} \sup_{||r||_\infty \leq 1} \mathbb{E}_\pi r - \mathbb{E}_{\pi_{\mathrm{ref}}} r \leq \sqrt{\frac{1}{2}\mathsf{KL}(\pi||\pi_{\mathrm{ref}})}$. Hence we can deduce that for bounded rewards $r$ with norm infinity $||r||_\infty$ that:

$$\mathbb{E}_\pi r - \mathbb{E}_{\pi_{\mathrm{ref}}} r \leq \sqrt{2||r||_\infty^2 \mathsf{KL}(\pi||\pi_{\mathrm{ref}})}.$$

Nevertheless this boundedness assumption on the reward is not realistic, since most reward models are unbounded: quoting Lambert et al. (2024b) "implemented by appending a linear layer to predict one logit or removing the final decoding layers and replacing them with a linear layer" and hence the reward is unbounded by construction. We will show in what follows that those scalings laws are tied to the tails of the reward under the reference policy and are instances of transportation inequalities. Note that the reward can be rescaled and transformed to be bounded, nevertheless Pinsker inequality remains loose, as with transportation inequality we aim at replacing the $||r||_\infty$, by a second moment or a standard deviation $\sigma$, and typically if $\sigma \ll ||r||_\infty$ this leads to tigther bounds.

### 3.1 Transportation Inequalities with $\mathsf{KL}$ Divergence

For a policy $\pi \in \mathcal{P}(\mathcal{Y})$ and for a reward function $r : \mathcal{Y} \to \mathbb{R}$ , we note $r_\sharp \pi$, the push-forward map of $\pi$ through $r$. The reader is referred to Appendix D.1 for background on transportation inequalities and how they are derived from the so-called Donsker-Varadhan variational representation of the $\mathsf{KL}$ divergence. The following Proposition hinges on Lemma 4.14 in (Boucheron et al., 2013)):

**Proposition 3 (Transportation Inequalities)** *The following inequalities hold depending on the tails of* $r_\sharp \pi_{\mathrm{ref}}$:

1. *Assume that $r_\sharp \pi_{\mathrm{ref}} \in \mathsf{SubGauss}(\sigma_{\mathrm{ref}}^2)$. For any $\pi \in \mathcal{P}(\mathcal{Y})$ that is absolutely continuous with respect to $\pi_{\mathrm{ref}}$, and such that $\mathsf{KL}(\pi||\pi_{\mathrm{ref}}) < \infty$ then we have:*

$$|\mathbb{E}_\pi r - \mathbb{E}_{\pi_{\mathrm{ref}}} r| \leq \sqrt{2\sigma_{\mathrm{ref}}^2 \mathsf{KL}(\pi||\pi_{\mathrm{ref}})}.$$

2. *Assume that $r_\sharp \pi_{\mathrm{ref}} \in \mathsf{SubGamma}(\sigma_{\mathrm{ref}}^2, c)$. For any $\pi \in \mathcal{P}(\mathcal{Y})$ that is absolutely continuous with respect to $\pi_{\mathrm{ref}}$, and such that $\mathsf{KL}(\pi||\pi_{\mathrm{ref}}) < \infty$ then we have:*

$$|\mathbb{E}_\pi r - \mathbb{E}_{\pi_{\mathrm{ref}}} r| \leq \sqrt{2\sigma_{\mathrm{ref}}^2 \mathsf{KL}(\pi||\pi_{\mathrm{ref}})} + c\mathsf{KL}(\pi||\pi_{\mathrm{ref}})$$

In particular we have the following Corollary:

**Corollary 1 (Expected Reward Improvement)** *If $r_\sharp \pi_{\mathrm{ref}} \in \mathsf{SubGauss}(\sigma_{\mathrm{ref}}^2)$ the following holds for the optimal RL policy $\pi_{\lambda_\Delta, r}$ and for the best of n policy $\pi_{r,\mathrm{ref}}^{(n)}$:*

1. *For the optimal RL policy $\pi_{\lambda_\Delta, r}$ we have:*

$$0 \leq \mathbb{E}_{\pi_{\lambda_\Delta, r}} r - \mathbb{E}_{\pi_{\mathrm{ref}}} r \leq \sqrt{2\sigma_{\mathrm{ref}}^2 \mathsf{KL}(\pi_{\lambda_\Delta, r}||\pi_{\mathrm{ref}})} \leq \sqrt{2\sigma_{\mathrm{ref}}^2 \Delta}.$$

2. *For any feasible policy $\pi_{\mathrm{RL}}$ of the RL problem equation 1 (for example the one obtained via gradient descent), we have:*

$$|\mathbb{E}_{\pi_{\mathrm{RL}}} r - \mathbb{E}_{\pi_{\mathrm{ref}}} r| \leq \sqrt{2\sigma_{\mathrm{ref}}^2 \mathsf{KL}(\pi_{\mathrm{RL}}||\pi_{\mathrm{ref}})} \leq \sqrt{2\sigma_{\mathrm{ref}}^2 \Delta}$$

3. *For the Best of n policy $\pi_{r,\mathrm{ref}}^{(n)}$, under Assumption 2 we have:*

$$0 \leq \mathbb{E}_{\pi_{r,\mathrm{ref}}^{(n)}} r - \mathbb{E}_{\pi_{\mathrm{ref}}} r \leq \sqrt{2\sigma_{\mathrm{ref}}^2 \mathsf{KL}(\pi_{r,\mathrm{ref}}^{(n)}||\pi_{\mathrm{ref}})} \leq \sqrt{2\sigma_{\mathrm{ref}}^2 \left(\log n - \frac{n-1}{n}\right)}$$

A similar statement holds under sub-gamma tails of the reward of the reference model. Item (1) in Corollary 1 shows that the $\sqrt{\sigma_{\mathrm{ref}}^2 \mathsf{KL}}$ provides an upper bound on the reward improvement of the alignment under subgaussian tails of the reference reward. Under subgaussian tails of the reference, this information theoretic barrier cannot be broken with a better algorithm. On way to improve on the $\sqrt{\mathsf{KL}}$ ceiling is by aiming at having a reference model with a reward that has subgamma tails to improve the upper limit to $\sqrt{\sigma_{\mathrm{ref}}^2 \mathsf{KL}} + c\mathsf{KL}$, or to subexponential tails to be linear in the $\mathsf{KL}$. Item (2) can be seen as a refinement on the classical $\sqrt{2\sigma_{\mathrm{ref}}^2 \log(n)}$ upper bound on the expectation of maximum of subgaussians see for e.g Corollary 2.6 in (Boucheron et al., 2013). If in addition $r$ is positive and for $X = r_\sharp \pi_{\mathrm{ref}} - \mathbb{E}_{\pi_{\mathrm{ref}}} r$ we have for $t > 0$, $\mathbb{P}(X > t) \geq \mathbb{P}(|g| > t)$, where $g \sim \mathcal{N}(0, \sigma_\ell^2)$ (where $\sigma_\ell^2$ is a variance), then we have a matching lower bound for $\pi_{r,\mathrm{ref}}^{(n)}$ that scales with $\sqrt{\sigma_\ell^2 \log(n)}$ for sufficiently large $n$ (See (Kamath, 2015)).

**Remark 1 (Tightness of $\mathsf{KL}$ Transportation Inequality)** *If we assume that $r$ is one to one, and that rewards are gaussian with same variance $\sigma^2$: $r_\sharp \pi_{\mathrm{ref}} \sim \mathcal{N}(\mu, \sigma^2)$ and $r_\sharp \pi_{\mathrm{ref}} \sim \mathcal{N}(\mu', \sigma^2)$, then we have $\mathsf{KL}(\pi||\pi_{\mathrm{ref}}) = \mathsf{KL}(r_\sharp \pi||r_\sharp \pi_{\mathrm{ref}}) = \frac{(\mu - \mu')^2}{2\sigma^2}$. Assume $\mu \geq \mu'$, in that case we have: $\mathbb{E}_\pi r - \mathbb{E}_{\pi_{\mathrm{ref}}} r = \mu - \mu' = \sqrt{2\sigma^2 \mathsf{KL}(\pi||\pi_{\mathrm{ref}})}$, and the inequality is tight in this case.*

We turn now to providing a bound in high probability on the empirical reward improvement of RL. The following Theorem gives high probability bounds for the excess reward when estimated from empirical samples:

**Theorem 2 (High Probability Empirical Reward Improvement For RL)** *Assume $r_\sharp \pi_{\mathrm{ref}} \in \mathsf{SubGauss}(\sigma_{\mathrm{ref}}^2)$. Let $\gamma > 1$ and $t_0 > 0$. Let $\pi_{\gamma, r}$ be the optimal policy of the penalized RL problem given in Equation equation 3. Let $R_{i,\gamma}$ and $R_{i,\mathrm{ref}}, i = 1 \ldots m$ be the rewards evaluated at $m$ samples from*

$\pi_{\gamma,r}$ and $\pi_{\text{ref}}$. *Assume that the $\gamma$-Rényi divergence $D_\gamma(\pi_{\gamma,r}||\pi_{\text{ref}})$ and $\mathsf{KL}(\pi_{\gamma,r}||\pi_{\text{ref}})$ are both finite. The following inequality holds with probability at least $1 - e^{-\frac{mt_0^2}{2\sigma_{\text{ref}}^2}} - e^{-m(\gamma-1)t_0}$:*

$$\frac{1}{m}\sum_{i=1}^{m} R_{i,\gamma} - \frac{1}{m}\sum_{i=1}^{m} R_{i,\text{ref}} \le \sqrt{2\sigma_{\text{ref}}^2 \mathsf{KL}(\pi_{\gamma,r}||\pi_{\text{ref}})} + \frac{D_\gamma(\pi_{\gamma,r}||\pi_{\text{ref}}) - \mathsf{KL}(\pi_{\gamma,r}||\pi_{\text{ref}})}{\gamma} + 2t_0.$$

Note that in Theorem 2, we did not make any assumptions on the tails of $r_\sharp \pi_{\gamma,r}$ and we see that this results in a biased concentration inequality with a non-negative bias $\frac{D_\gamma(\pi_{\gamma,r}||\pi_{\text{ref}}) - \mathsf{KL}(\pi_{\gamma,r}||\pi_{\text{ref}})}{\gamma} \ge 0$. For the best of $n$ policy, if the reward was positive and has a folded normal distribution (absolute value of gaussians), Boucheron & Thomas (2012) provides concentration bounds, owing to subgamma tails of maximum of absolute Gaussians.

### 3.2 Tail Adaptive Transportation Inequalities with the Rényi Divergence

An important question on the tightness of the bounds rises from the bounds in Corollary 1. We answer this question by considering additional information on the tails of the reward under the policy $\pi$, and we obtain tail adaptive bounds that are eventually tighter than the one in Corollary 1. Our new bounds leverage a variational representation of the Rényi divergence that uses the logarithmic moment generating function of both measures at hand.

**Preliminaries for the Rényi Divergence** The Donsker-Varadahn representation of $\mathsf{KL}$ was crucial in deriving transportation inequalities. In (Shayevitz, 2011) the following variational form is given for the Rényi divergence in terms of the $\mathsf{KL}$ divergence, for all $\alpha \in \mathbb{R}$

$$(1-\alpha)D_\alpha(P||Q) = \inf_R \alpha\mathsf{KL}(R||P) + (1-\alpha)\mathsf{KL}(R||Q) \tag{15}$$

A similar variational form was rediscovered in (Anantharam, 2018). Finally a Donsker-Varadahn-Rényi representation of $D_\alpha$ was given in (Birrell et al., 2021). For all $\alpha \in \mathbb{R}^+, \alpha \neq 0, 1$ we have :

$$\frac{1}{\alpha}D_\alpha(P||Q) = \sup_{h \in \mathcal{H}} \frac{1}{\alpha-1}\log\left(\mathbb{E}_P e^{(\alpha-1)h}\right) - \frac{1}{\alpha}\log\left(\mathbb{E}_Q e^{\alpha h}\right), \tag{16}$$

where $\mathcal{H} = \left\{h \middle| \int e^{(\alpha-1)h}dP < \infty, \int e^{\alpha h}dQ < \infty\right\}$. Birrell et al. (2021) presents a direct proof of this formulation without exploring its link to the representation given in equation 15, we show in what follows an elementary proof via convex conjugacy, the duality relationship between equations equation 15 and equation 16.

**Theorem 3** *For $0 < \alpha < 1$ Equations equation 15 and equation 16 are dual of one another. For $\alpha > 1$ they are Toland Dual.*

We collect in what follows elementary lemmas that will be instrumental to derive transportation inequalities in terms of the Rényi divergence. Proofs are given in the Appendix.

**Lemma 2** *Let $\alpha \in (0,1) \cup (1,\infty)$, and define $\mathcal{H} = \{h | e^{(\alpha-1)(h-\int hdP)} \in L^1(P), e^{(\alpha)(h-\int hdQ)} \in L^1(Q)\}$. We have for all $h \in \mathcal{H}$ and for $\alpha \in (0,1) \cup (1,\infty)$*

$$\int hdP - \int hdQ \le \frac{1}{\alpha}D_\alpha(P||Q) - \frac{1}{\alpha-1}\log\left(\int e^{(\alpha-1)(h-\int hdP)}dP\right) + \frac{1}{\alpha}\log\left(\int e^{\alpha(h-\int hdQ)}dQ\right).$$

**Lemma 3** *The following limit holds for the Rényi divergence $\lim_{\alpha\to 0} \frac{1}{\alpha}D_\alpha(P||Q) = \mathsf{KL}(Q||P)$.*

**Transportation Inequalities with Rényi Divergence.** The following theorem shows that when considering the tails of $\pi$ we can obtain tighter upper bounds using the Rényi divergence that is more tail adaptive:

**Theorem 4 (Tail Adaptive Transportation Inequalities)** *Let $\alpha \in (0,1)$. Assume $r_\sharp \pi \in \mathsf{SubGauss}(\sigma_\pi^2)$ and $r_\sharp \pi_{\mathrm{ref}} \in \mathsf{SubGauss}(\sigma_{\mathrm{ref}}^2)$ then we have for all $\alpha \in (0,1)$:*

$$\mathbb{E}_\pi r - \mathbb{E}_{\pi_{\mathrm{ref}}} r \leq \sqrt{2((1-\alpha)\sigma_\pi^2 + \alpha \sigma_{\mathrm{ref}}^2)\frac{D_\alpha(\pi||\pi_{\mathrm{ref}})}{\alpha}}. \tag{17}$$

In particular if there exits $\alpha \in (0,1)$ such that $D_\alpha(\pi||\pi_{\mathrm{ref}}) \leq \frac{\alpha \sigma_{\mathrm{ref}}^2}{(1-\alpha)\sigma_\pi^2 + \alpha \sigma_{\mathrm{ref}}^2}\mathsf{KL}(\pi||\pi_{\mathrm{ref}})$, then the tail adaptive upper bound given in Equation equation 17 is tighter than the one provided by the tails of $\pi_{\mathrm{ref}}$ only i.e $\sqrt{\sigma_{\mathrm{ref}}^2 \mathsf{KL}(\pi||\pi_{\mathrm{ref}})}$. Note that this is possible because $D_\alpha$ is increasing in $\alpha \in (0,1)$ (van Erven & Harremos, 2014), i.e $D_\alpha(\pi||\pi_{\mathrm{ref}}) \leq \mathsf{KL}(\pi||\pi_{\mathrm{ref}})$, and $\frac{\alpha \sigma_{\mathrm{ref}}^2}{(1-\alpha)\sigma_\pi^2 + \alpha \sigma_{\mathrm{ref}}^2} \leq 1$. Note that taking limits $\alpha \to 0$ (applying Lemma 3) and $\alpha \to 1$, and taking the minimum of the upper bounds we obtain:

$$\mathbb{E}_\pi r - \mathbb{E}_{\pi_{\mathrm{ref}}} r \leq \sqrt{2\min(\sigma_{\pi_{\mathrm{ref}}}^2 \mathsf{KL}(\pi||\pi_{\mathrm{ref}}), \sigma_\pi^2 \mathsf{KL}(\pi_{\mathrm{ref}}||\pi))},$$

this inequality can be also obtained by applying Proposition 3 twice: on the tails of $\pi$ and $\pi_{\mathrm{ref}}$ respectively.

Another important implication of Theorem 4, other than tighter than $\mathsf{KL}$ upper bound, is that if we were to change the RL alignment problem equation 1 to be constrained by $D_\alpha, \alpha \in (0,1)$ instead of $\mathsf{KL}$, we may end up with a smaller upper limit on the reward improvement. This $D_\alpha$ constrained alignment may lead to a policy that under-performs when compared to a policy obtained with the $\mathsf{KL}$ constraint. This was indeed observed experimentally in (Wang et al., 2024a) that used constraints with $\alpha$- divergences for $\alpha \in (0,1)$ (that are related to Rényi divergences) and noticed a degradation in the reward improvement w.r.t policies obtained using $\mathsf{KL}$.

### 3.3 TV-**Transportation Inequalities for Bounded and Calibrated Rewards**

While in the previous sections we focused on potentially unbounded rewards and analyzed their tail behavior to derive transportation inequalities linking rewards to $\mathsf{KL}$ or Rényi divergences, in this Section, we turn to establishing sharp transportation inequalities for the best of $n$ policy under bounded rewards. Such rewards may be inherently bounded, as in the case of win rates (Azar et al., 2024), or they may be obtained through transformation or calibration. For instance, one approach is to use the cumulative distribution function (CDF) of the reward under the reference model, $F_{r,\pi_{\mathrm{ref}}}$, to calibrate the reward of the best of $n$ policy (Beirami et al., 2024; Balashankar et al., 2024; Nitsure et al., 2024; Belgodere et al., 2024).

The total variation (TV) distance, being both an integral probability metric and an $f$-divergence, satisfies the data processing inequality. Under Assumption 2, as demonstrated in Table 1, we have:

$$\mathsf{TV}(\pi_{r,\mathrm{ref}}^{(n)}||\pi_{\mathrm{ref}}) = \frac{1}{2}\sup_{||r||_\infty \leq 1} \mathbb{E}_{\pi_{r,\mathrm{ref}}^{(n)}} r - \mathbb{E}_{\pi_{\mathrm{ref}}} r \leq \mathsf{TV}(E^{(n)}||E) = \left(\frac{1}{n}\right)^{\frac{1}{n-1}} - \left(\frac{1}{n}\right)^{\frac{n}{n-1}}.$$

Hence, for any bounded reward:

$$\mathbb{E}_{\pi_{r,\mathrm{ref}}^{(n)}} r - \mathbb{E}_{\pi_{\mathrm{ref}}} r \leq 2\left\|r\right\|_\infty \left(\left(\frac{1}{n}\right)^{\frac{1}{n-1}} - \left(\frac{1}{n}\right)^{\frac{n}{n-1}}\right).$$

We note that our bounds are translation-invariant. If $r \in [0,1]$, defining $\tilde{r} = r - \frac{1}{2}$ such that $\tilde{r} \in [-\frac{1}{2}, \frac{1}{2}]$ and $\|\tilde{r}\|_\infty = \frac{1}{2}$ , we obtain:

$$\mathbb{E}_{\pi_{r,\mathrm{ref}}^{(n)}} r - \mathbb{E}_{\pi_{\mathrm{ref}}} r = \mathbb{E}_{\pi_{\tilde{r},\mathrm{ref}}^{(n)}} \tilde{r} - \mathbb{E}_{\pi_{\mathrm{ref}}} \tilde{r} \leq 2\left\|\tilde{r}\right\|_\infty \left(\left(\frac{1}{n}\right)^{\frac{1}{n-1}} - \left(\frac{1}{n}\right)^{\frac{n}{n-1}}\right) = \left(\frac{1}{n}\right)^{\frac{1}{n-1}} - \left(\frac{1}{n}\right)^{\frac{n}{n-1}}.$$

In particular, by calibrating the reward using $F_{r,\pi_{\text{ref}}} \circ r$, we obtain a bounded reward where $(F_{r,\pi_{\text{ref}}} \circ r)_{\sharp}\pi_{\text{ref}}$ is uniform. Since $\mathbb{E}_{\pi_{\text{ref}}}(F_{r,\pi_{\text{ref}}} \circ r) = \frac{1}{2}$ (by uniformity), we derive the following bound for the best of $n$ policy:

$$\mathbb{E}_{\pi_{r,\text{ref}}^{(n)}}(F_{r,\pi_{\text{ref}}} \circ r) - \frac{1}{2} \le \left(\frac{1}{n}\right)^{\frac{1}{n-1}} - \left(\frac{1}{n}\right)^{\frac{n}{n-1}} = \mathsf{TV}(E^{(n)}||E). \tag{18}$$

**Discussion**  The upper bound in equation 18 is exactly $\mathsf{TV}(E^{(n)}||E)$ and is tighter than what would be obtained using the subgaussianity of the calibrated reward. Since $(F_{r,\pi_{\text{ref}}} \circ r)_{\sharp}\pi_{\text{ref}} \in \mathsf{SubGauss}(\frac{1}{4})$, Corollary 1 (2) yields:

$$\mathbb{E}_{\pi_{r,\text{ref}}^{(n)}}(F_{r,\pi_{\text{ref}}} \circ r) - \frac{1}{2} \le \sqrt{\frac{1}{2}\left(\log(n) - \frac{n-1}{n}\right)} = \sqrt{\frac{1}{2}\mathsf{KL}(E^{(n)}||E)}. \tag{19}$$

Comparing equation 18 and equation 19, we see that $\mathsf{TV}$ provides a tighter upper bound. By Pinsker's inequality:

$$\mathsf{TV}(E^{(n)}||E) \le \sqrt{\frac{1}{2}\mathsf{KL}(E^{(n)}||E)},$$

which shows that equation 19 provides a looser bound. Moreover Pinkser inequality is known to be vaccuous whenerver $\mathsf{KL} > 1$, a tighter upper bound for $\mathsf{KL}$ can be derived using the Bretagnolle-Huber inequality Polyanskiy & Wu (2023):

$$\mathsf{TV}(E^{(n)}||E) \le \sqrt{1 - e^{-\mathsf{KL}(E^{(n)}||E)}}.$$

Substituting $\mathsf{KL}(E^{(n)}||E)$ from Lemma 1 into equation 18, we obtain:

$$\mathbb{E}_{\pi_{r,\text{ref}}^{(n)}}(F_{r,\pi_{\text{ref}}} \circ r) - \frac{1}{2} \le \left(\frac{1}{n}\right)^{\frac{1}{n-1}} - \left(\frac{1}{n}\right)^{\frac{n}{n-1}} \le \sqrt{1 - e^{-\log(n) + \frac{n-1}{n}}} = \sqrt{1 - \frac{1}{n}e^{\frac{n-1}{n}}}. \tag{20}$$

We illustrate these bounds in Figure 6 for uniform and beta distributed random variables. We see that $\mathsf{TV}$ provides the sharpest upper bound followed by the Bretagnolle-Huber's upper bound and finally by the $\mathsf{KL}$ upper bound.

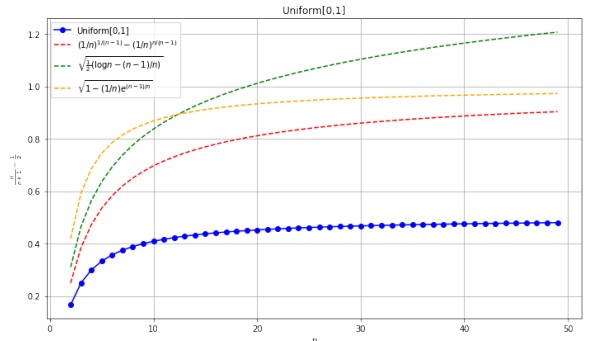

(a) Bounds for uniform random variable.

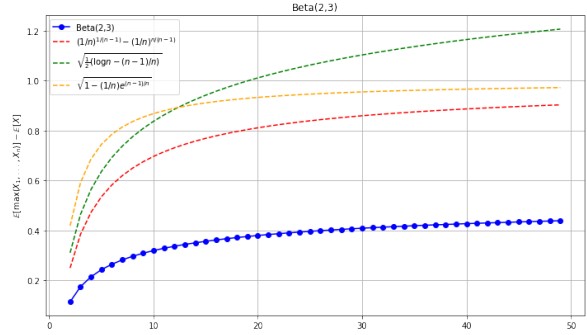

(b) Bounds for Beta distributed random variable

Figure 6: Information Theoretic Upper Bounds for Best of $n$ in the bounded case.

## 4 Transportation Inequality Transfer From Proxy to Golden Reward

As we saw in the previous sections, the tightness of $\sqrt{\mathsf{KL}(\pi||\pi_{\text{ref}})}$ upper bound in alignment can be due to the tails of the reward of the aligned policy $\pi$ (Theorem 4) and to the concentration around the mean in finite sample size (Theorem 2). Another important consideration is the mismatch between the golden reward $r^*$ that one desires to maximize that is expensive and difficult to obtain (for example human evaluation) and a proxy reward $r$ that approximates $r^*$. The proxy reward $r$ is used instead of $r^*$ in RL and in best of

$n$ policy. While we may know the tails of the reward $r$ of the reference and aligned model, we don't have access to this information on the golden reward $r^*$. We show in this section how to transfer transportation inequalities from $r$ to $r^*$ for RL and Best of $n$ policy.

**Proposition 4 ($r^*$ Transportation Inequality for RL Policy )** *The following inequality holds:*

$$\mathbb{E}_{\pi_{\gamma,r}} r^* - \mathbb{E}_{\pi_{\text{ref}}} r^* \leq \mathbb{E}_{\pi_{\gamma,r}} r - \mathbb{E}_{\pi_{\text{ref}}} r - \frac{1}{\gamma} \log \left( \int e^{\gamma(r - r^* - \left( \int r d\pi_{\text{ref}} - \int r^* d\pi_{\text{ref}} \right))} d\pi_{\gamma,r^*} \right),$$

*Assume* $r_\sharp \pi_{\text{ref}} \in \mathsf{SubGauss}(\sigma_{\text{ref}}^2)$, *and there exists* $\delta > 0$ *such that:* $\frac{1}{\gamma} \log \left( \int e^{\gamma(r - r^* - \left( \int r d\pi_{\text{ref}} - \int r^* d\pi_{\text{ref}} \right))} d\pi_{\gamma,r^*} \right) \geq \delta \mathsf{KL}(\pi_{\gamma,r^*} || \pi_{\text{ref}})$, *then we have:*

$$\mathbb{E}_{\pi_{\gamma,r}} r^* - \mathbb{E}_{\pi_{\text{ref}}} r^* \leq \sqrt{2\sigma_{\text{ref}}^2 \mathsf{KL}(\pi_{\gamma,r} || \pi_{\text{ref}})} - \delta \mathsf{KL}(\pi_{\gamma,r^*} || \pi_{\text{ref}}).$$

Note that $\frac{1}{\gamma} \log \left( \int e^{\gamma(r - r^* - \left( \int r d\pi_{\text{ref}} - \int r^* d\pi_{\text{ref}} \right))} d\pi_{\gamma,r^*} \right)$ is interpreted here as an interpolation between the mean and the maximum of its argument on the support of $\pi_{\gamma,r^*}$ (Proposition 9 in (Feydy et al., 2018)). Indeed as $\gamma \to 0$, this boils down to the mean on $\int (r - r^*) d\pi_{\gamma,r^*} - \left( \int r d\pi_{\text{ref}} - \int r^* d\pi_{\text{ref}} \right)$ and $\gamma \to \infty$ this boils down to $\max_{\mathsf{supp}\pi_{\gamma,r^*}} \{ r - r^* - \left( \int r d\pi_{\text{ref}} - \int r^* d\pi_{\text{ref}} \right) \}$. Our assumption means that $r$ overestimates $r^*$ and the overestimation is accentuated as we drift from $\pi_{\text{ref}}$ on which $r$ was learned. If $r$ overestimates $r^*$, there exists $\Delta > 0$ such that: $r - r^* - \left( \int r d\pi_{\text{ref}} - \int r^* d\pi_{\text{ref}} \right) \geq \Delta$ By Jensen inequality we have: $\frac{1}{\gamma} \log \left( \int e^{\gamma(r - r^* - \left( \int r d\pi_{\text{ref}} - \int r^* d\pi_{\text{ref}} \right))} d\pi_{\gamma,r^*} \right) \geq \frac{1}{\gamma} \int \gamma(r - r^* - \left( \int r d\pi_{\text{ref}} - \int r^* d\pi_{\text{ref}} \right)) d\pi_{\gamma,r^*} \geq \Delta$. Hence our assumption is on the overestimation error $\Delta = \delta \mathsf{KL}(\pi_{\gamma,r^*} || \pi_{\text{ref}})$. This assumption echoes findings in (Gao et al., 2023) that show that the transportation inequalities suffer from overestimation of proxy reward models of the golden reward (See Figure 8 in (Gao et al., 2023)).

Note that in Proposition 4, we are evaluating the golden reward $r^*$ improvement when using the proxy reward optimal policy $\pi_{\gamma,r}$. We see that the golden reward of the RL policy inherits the transportation inequality from the proxy one but the improvement of the reward is hindered by possible overestimation of the golden reward by the proxy model. This explains the dip in performance as measured by the golden reward depicted in Figure 1 and reported in (Gao et al., 2023).

**Proposition 5 ($r^*$ Transportation Inequality for Best of $n$ Policy)** *Let $\varepsilon > 0$. Let $r$ be a surrogate reward such that $\|r - r^*\|_\infty \leq \varepsilon$ and assume $r_\sharp \pi_{\text{ref}} \in \mathsf{SubGauss}(\sigma_{\text{ref}}^2)$ then the best of $n$ policy $\pi_{r,\text{ref}}^{(n)}$ satisfies:*

$$\mathbb{E}_{\pi_{r,\text{ref}}^{(n)}}(r^*) - \mathbb{E}_{\pi_{\text{ref}}}(r^*) \leq \sqrt{2\sigma_{\text{ref}}^2 \left( \log(n) - \frac{n-1}{n} \right)} + 2\varepsilon \left( \left( \frac{1}{n} \right)^{\frac{1}{n-1}} - \left( \frac{1}{n} \right)^{\frac{n}{n-1}} \right).$$

Transportation inequalities transfers for the best of $n$ policy from $r$ to $r^*$ and pays only an additional error term $\|r - r^*\|_\infty \mathsf{TV}(\pi_{r,\text{ref}}^{(n)} | \pi_{\text{ref}})$ , an upper bound of this total variation as a function of $n$ is given in Table 1. As mentioned earlier, if we have lower bounds on the tail of the reference reward, then we also have a lower bound on the reward improvement that scales like $C\sqrt{\sigma_\ell^2 \log(n)} - 2\varepsilon \left( \left( \frac{1}{n} \right)^{\frac{1}{n-1}} - \left( \frac{1}{n} \right)^{\frac{n}{n-1}} \right)$. This is in line with empirical findings in (Hilton & Gao, 2022) (Gao et al., 2023) that showed that best of $n$ policy is resilient as the reward model $r$ gets closer to $r^*$.

**Practical Implications.** As the proxy reward may overestimate the golden reward, (Wang et al., 2024b) proposed a reward transformation that reduces the tails of the reward distribution. This, in turn, improves the golden reward by avoiding shortcuts that exploit the reward and target the unreliable tails of the distribution, which overestimate the golden reward with high values. In our theory, this would reduce $\delta$ in Proposition 4, leading to less catastrophic Goodhart.

## 5 Numerical Results

**Prompts Dataset, LLMs and Reward Models** We consider the attaq dataset (Kour et al., 2023) consisting of 1.4k prompts that triggers undesirable behaviors in LLMs. We use as Reward model FsFAIRX-

LLAMA3-RM-v0.1 (Dong et al., 2023; Xiong et al., 2024) this is among the best reward model for measuring helpfulness, safety, instruction following and lack of toxicity (Lambert et al., 2024a). We use three LLMs from each we sample with top-p sampling and a temperature $\tau$ 100 responses for each prompt, models are: MERLINITE (Sudalairaj et al., 2024) (a base model not aligned), MIXTRAL-8X7B-INSTRUCT (Jiang et al., 2024) and LLAMA-2-13B-CHAT (Touvron et al., 2023) (aligned models with different reward).

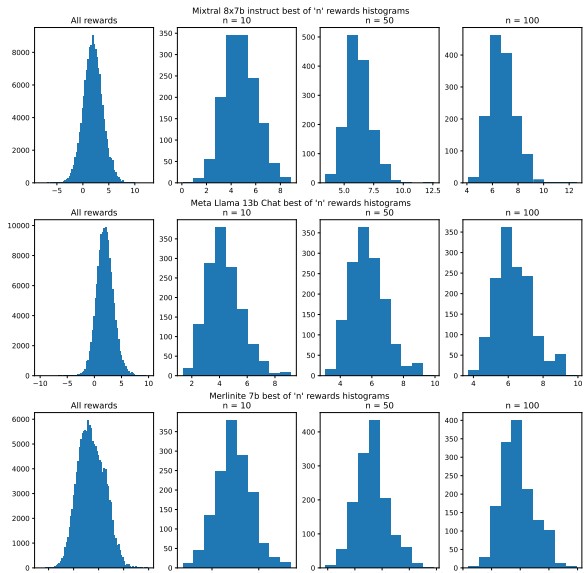

Figure 7: Histogram of reward and best of $n$ reward with FSFAIRX-LLAMA3-RM-v0.1

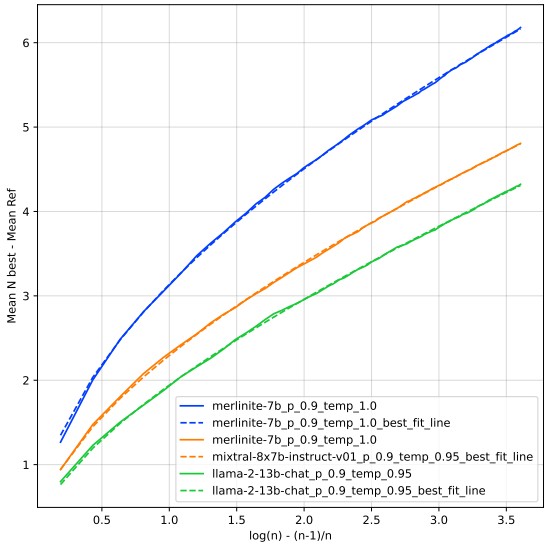

Figure 8: (y-axis) The centered reward using FSFAIRX-LLAMA3-RM-v0.1 reward model of best of $n$ policy $\mathbb{E}_{\pi_{r,\mathrm{ref}}^n} r - \mathbb{E}_{\pi_{\mathrm{ref}}} r$ versus (x-axis) $\mathsf{KL}(\pi_{r,\mathrm{ref}}^n, \pi_{\mathrm{ref}})$ policy computed as $\log(n) - \frac{n-1}{n}$. We also plot fitted curves $y = a\sqrt{x} + bx$. Coefficients are given in Tab. 3.

**Results and Discussion** We plot in Figure 7 the histograms of the reward under the original LLM policy (first panel) and under the best of $n$ policy for $n = 10, 50, 100$ (second to fourth panels). We observe that the reference rewards are not heavy-tailed and follow sub-Gaussian/sub-Gamma distributions ( appendix G provides tests such as q-q plots to probe this). Table 2 presents statistics of these distributions. We see that

Mixtral achieves the highest mean reward using the best of $n$ for $n = 100$, while Merlinite shows the greatest improvement over the original policy.

This is due to the fact that Merlinite is a base model and not an aligned one. In Figure 8, we plot the centered best of $n$ rewards against the KL divergence of the best of $n$ policy to the reference policy, and observe that it follows the form $a\sqrt{x} + bx$, indicating a sub-Gamma tail as shown in Proposition 3. Looking at the estimated coefficient $a$ in Table 3 and the standard deviation of the reward under reference models in Table 2 (denoted as all), we observe that $a$ is slightly larger than the standard deviation of the reward under the reference policy, which aligns with our theory. If we apply Pinsker's inequality, the upper bound on the reward would scale with the maximum reward, as also given in Table 2 , which is much larger than $a$, resulting in a loose bound. Thus, the reward/KL plots are governed by the tail properties of the reference policy.

| Model and best of n | Mean | Std | Max |
|---|---|---|---|
| MERLINITE ALL | -1.42 | 2.66 | 9.91 |
| MERLINITE n = 10 | 2.33 | 1.81 | 8.44 |
| MERLINITE n = 50 | 4.07 | 1.45 | 9.74 |
| MERLINITE n = 100 | 4.76 | 1.40 | 9.91 |
| MIXTRAL ALL | 1.96 | 1.86 | 12.57 |
| MIXTRAL n = 10 | 4.74 | 1.31 | 8.86 |
| MIXTRAL n = 50 | 6.21 | 1.03 | 12.57 |
| MIXTRAL n = 100 | 6.78 | 0.98 | 12.57 |
| LLAMA13BCHAT ALL | 1.98 | 1.58 | 9.92 |
| LLAMA13BCHAT n = 10 | 4.35 | 1.23 | 9.12 |
| LLAMA13BCHAT n = 50 | 5.74 | 1.09 | 9.92 |
| LLAMA13BCHAT n = 100 | 6.31 | 1.04 | 9.92 |

Table 2: Comparison of Mean, Std, and Max values for the reward model FSFAIRX-LLAMA3-RM-V0.1 evaluating samples from different models using various best of $n$ policies.

| Model | a | b |
|---|---|---|
| Merlinite-7B | 3.01309273 | 0.12227016 |
| Mixtral-8x7B-instruct | 2.03057845 | 0.26265606 |
| Llama 13B-chat | 1.57822599 | 0.36367755 |

Table 3: Best fitted curves coefficients $y = a\sqrt{x} + bx$

**Reward Versus Rényi divergence for Best of** $n$    Varying $n$, we compute the centered expected reward of the best of $n$ policy versus the $\alpha$ Rényi divergence. The Rényi divergence of the best of $n$ policy is computed using the expression given in Table 1. For $\alpha \in (0, 1)$ for smaller values of the Rényi divergence we achieve higher reward than KL (See Figures 22, 23 and 24). For $\alpha > 1$, for same value of KL, we achieve higher reward using KL than using the Rényi divergence for $\alpha > 1$, (See Figures 25, 26 and 27). This suggests that for same value of the divergence, $\alpha$-Rényi divergence for $\alpha > 1$ may allow for less reward hacking. This observation was used in (Huang et al., 2024) using the Chi-Squared regularizer ($\alpha = 2$) in addition to KL to fight reward overoptimization.

**Calibrated Reward versus** TV **and** KL    We follow the notations of Section 3.3 for centered calibrated rewards using the CDF $F_{r,\pi_{\text{ref}}}$ of the reward under the reference model. In Figure 9, we plot the centered calibrated rewards of the best of $n$ policy for the three LLMs considered using the reward model FSFAIRX-LLAMA3-RM-V0.1 against (a) the TV upper bound given in equation 18, (b) the $\sqrt{\frac{1}{2}\text{KL}}$ upper bound given in equation 19, and (c) the $\sqrt{1 - e^{-\text{KL}}}$ upper bound given in equation 20.

From Figure 9, we observe that (a) TV provides the sharpest upper bound, showing a linear correspondence between the centered calibrated rewards of the best of $n$ policy and TV. The next tightest bound is (c)

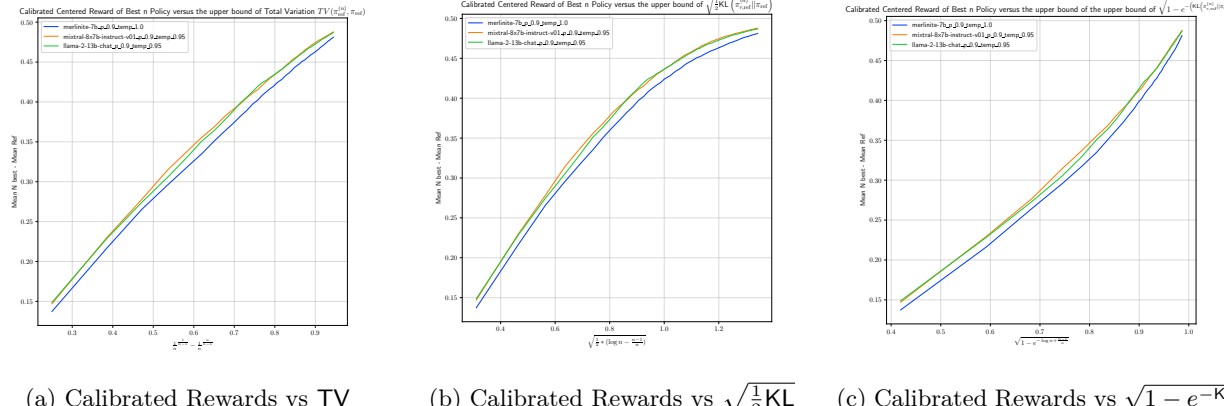

(a) Calibrated Rewards vs $\mathsf{TV}$       (b) Calibrated Rewards vs $\sqrt{\frac{1}{2}\mathsf{KL}}$       (c) Calibrated Rewards vs $\sqrt{1-e^{-\mathsf{KL}}}$

Figure 9: (y-axis) The Calibrated centered reward using FSFAIRX-LLAMA3-RM-v0.1 reward model of best of $n$ policy $\mathbb{E}_{\pi_{r,\mathrm{ref}}^n} F_{r,\pi_{\mathrm{ref}}} \circ r - \frac{1}{2}$ versus on (x-axis) (a) $\mathsf{TV}$ upper bound given in equation 18, (b) $\sqrt{\frac{1}{2}\mathsf{KL}}$ upper bound given equation 19 (c) $\sqrt{1-e^{-\mathsf{KL}}}$ upper bound given in equation 20.

$\sqrt{1-e^{-\mathsf{KL}}}$, derived via the Bretagnolle-Huber inequality, followed by (b) $\sqrt{\frac{1}{2}\mathsf{KL}}$, obtained via Pinsker's inequality, which, as expected, is the loosest among them.

## 6 Conclusion

We presented in this paper a comprehensive information theoretical analysis of policy alignment using reward optimization with RL and best of $n$ sampling. We showed for best of $n$ a bound on $\mathsf{KL}$ under realistic assumptions on the reward. Our analysis showed that the alignment reward improvement, is intrinsically constrained by the tails of the reward under the reference policy and controlling the $\mathsf{KL}$ divergence results in an upper bound of the policy improvement. We showed that the $\mathsf{KL}$ bound may not be tight if the tails of the optimized policy satisfy a condition expressed via Rényi divergence. We also explained the deterioration of the golden reward via overestimation of the proxy reward.

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

## A    Broader Impact and Limitations

We believe this work explaining scaling laws for reward models and alignment will give practitioners insights regarding the limits of what is attainable via alignment. All assumptions under which our statements hold are given. We don't see any negative societal impact of our work.

## B    Proofs For Best of $n$ Policy

### B.1    Best of $n$ Policy KL Guarantees

**Proof 1 (Proof of Lemma 1)**

$$\mathsf{KL}(E^{(n)}||E) = \int_0^{+\infty} f_{E^{(n)}}(x) \log\left(\frac{f_{E^{(n)}}(x)}{f_E(x)}\right) dx$$

We have $f_E(x) = e^{-x}1_{x\geq 0}$. Note that the CDF of maximum of exponential $F_{E^{(n)}}(x) = (1-e^{-x})^n 1_{x\geq 0}$, and hence $f_{E^{(n)}}(x) = n(1-e^{-x})^{n-1}e^{-x}1_{x\geq 0}$. Hence we have:

$$\mathsf{KL}(E^{(n)}||E) = \int_0^{+\infty} n(1-e^{-x})^{n-1}e^{-x} \log\left(\frac{n(1-e^{-x})^{n-1}e^{-x}}{e^{-x}}\right) dx$$

$$= \int_0^{+\infty} n(1-e^{-x})^{n-1}e^{-x} \log\left(n(1-e^{-x})^{n-1}\right) dx$$

Let $u = 1 - e^{-x}$, we have $du = e^{-x}dx$. It follows that :

$$\mathsf{KL}(E^{(n)}||E) = \int_0^1 nu^{n-1} \log\left(nu^{n-1}\right) du$$

$$= \int_0^1 nu^{n-1}\left(\log(n) + (n-1)\log(u)\right) du$$

$$= \log(n)\int_0^1 du^n + (n-1)\int_0^1 nu^{n-1}\log(u)du$$

$$= \log(n) + (n-1)\int_0^1 d(u^n \log u - \frac{u^n}{n})$$

$$= \log(n) - \frac{n-1}{n}.$$

**Proof 2 (Proof of Theorem 1)** *Recall that $T_X = F_{R(Y)|X}^{-1} \circ F_E$, $F_E$ is one to one. If the space $\mathcal{Y}$ is finite, $R(Y|X)$ has a discontinuous CDF hence not strictly monotonic. It follows that its quantile $F_{R(Y)|X}^{-1}$ is not a one to one map and $T_X$ as a result is not a one to one map and hence we have by DPI (that is an inequality in this case since $T_X$ is not one to one):*

$$\mathsf{KL}(T_X(E^{(n)})||T_X(E)) \leq \mathsf{KL}(E^{(n)}||E) \tag{21}$$

*If the space $\mathcal{Y}$ is infinite and we assume that $R(Y|X)$ is continuous and strictly monotonic then $F_{R(Y)|X}^{-1}$ is a one to one map, and as a result $T_X$ is a one to one map and the DPI is an equality in this case:*

$$\mathsf{KL}(T_X(E^{(n)})||T_X(E)) = \mathsf{KL}(E^{(n)}||E) \tag{22}$$

*Hence under Assumption 1 and for $\mathcal{Y}$ finite combining equation 11 and equation 21 we have:*

$$\mathsf{KL}(\pi_{r,\text{ref}}^{(n)}||\pi_{ref}|X) \leq \mathsf{KL}(E^{(n)}||E), \tag{23}$$

*and under Assumption 1 and for $\mathcal{Y}$ infinite and assuming $F_{R(Y)|X}$ is continuous and strictly monotonic, combining equation 11 and equation 22 we have:*

$$\mathsf{KL}(\pi_{r,\text{ref}}^{(n)}||\pi_{ref}|X) = \mathsf{KL}(E^{(n)}||E). \tag{24}$$

*Under the more realistic Assumption 2 we can also apply the DPI on the stochastic map $H_X$, since DPI also holds for stochastic maps ( under our assumption $R|X \to Y|X$ see for example (van Erven & Harremos, 2014) Example 2)*

$$\mathsf{KL}(\pi_{r,\text{ref}}^{(n)}||\pi_{ref}|X) = \mathsf{KL}(H_X(R^n(Y))||H_X(R(Y))|X))$$

$$\leq \mathsf{KL}(R^n(Y)||R(Y)|X) = \mathsf{KL}(T_X(E^{(n)})||T_X(E)), \tag{25}$$

*and hence under Assumption 2 regardless whether $T_X$ is a one to one map or not, thus we have:* $\mathsf{KL}(\pi_{r,\text{ref}}^{(n)}||\pi_{ref}|X) \leq \mathsf{KL}(E^{(n)}||E)$.

## B.2 Best of n Policy f divergence and Rényi Divergence

**Best of $n$ Policy $f$ divergence and Renyi divergence Guarantees**   Given that our proof technique relies on DPI and Rényi representation, we show that similar results hold for any $f$-divergence and for the Rényi divergence:

$$D_f(P||Q) = \int q(x) f\left(\frac{p(x)}{q(x)}\right) dx, \tag{26}$$

where $f$ is convex and $f(1) = 0$. Hence we have by DPI for $f$-divergences:

---

**Theorem 5**  *Under Assumption 2 the best of n policy satisfies for any f-divergence:*

$$\mathrm{D}_f(\pi_{r,\text{ref}}^{(n)}||\pi_{\text{ref}}) \leq \int_0^1 f\left(nu^{n-1}\right) du \tag{27}$$

---

**Proof 3 (Proof of Theorem 5)**

$$\mathrm{D}_f(\pi_{r,\text{ref}}^{(n)}||\pi_{ref}|X) = \mathrm{D}_f(Y^{(n)}||Y|X)$$

$$= \mathrm{D}_f(H_X(R^n(Y))||H_X(R(Y))|X)$$

$$\leq \mathrm{D}_f(R^n(Y)||R(Y)|X) \text{ By the data processing inequality} \tag{28}$$

$$= \mathrm{D}_f(T_X(E^{(n)})||T_X(E)) \text{ Renyi and Optimal Transport Representations equation 10}$$

$$= \mathrm{D}_f(E^{(n)}||E) \text{ since } T_X \text{ is a monotonic bijection DPI is an equality} \tag{29}$$

$$= \int_0^{+\infty} f_E(x) f\left(\frac{f_{E^{(n)}}(x)}{f_E(x)}\right) dx \tag{30}$$

$$= \int_0^{\infty} (e^{-x}) f\left(n(1 - e^{-x})^{n-1}\right) du \tag{31}$$

$$= \int_0^1 f(nu^{n-1}) du. \tag{32}$$

*In particular we have the following bounds for common f divergences:*

- *For $f(x) = x \log(x)$ we obtain the KL divergence and we have the result:*

$$\int_0^1 nu^{n-1} \log(nu^{n-1}) du = \mathsf{KL}(E^{(n)}||E) = \log(n) - \frac{n-1}{n}.$$

- *For $f(x) = (x-1)^2$ we obtain the chi-squared divergence and we have: $\int_0^1 \left(nu^{n-1} - 1\right)^2 du = \int_0^1 (n^2 u^{2(n-1)} - 2nu^{n-1} + 1) du = \frac{n^2}{2n-1} u^{2n-1} - 2u^n + u|_0^1 = \frac{n^2}{2n-1} - 2 + 1 = \frac{n^2 - 2n + 1}{2n-1} = \frac{(n-1)^2}{2n-1}$.*

- *For $f(x) = \frac{1}{2}|x-1|$, we obtain the total variation distance (TV) and we have: $\frac{1}{2}\int_0^1 |nu^{n-1} - 1|\, du = \frac{1}{2}(\int_0^{u^*}(1 - nu^{n-1})\, du + (\int_{u^*}^1 (nu^{n-1} - 1)\, du) = (u^* - (u^*)^n)$, where $n(u^*)^{(n-1)} = 1$, i.e $u^* = (\frac{1}{n})^{\frac{1}{n-1}}$. Hence the TV is $(\frac{1}{n})^{\frac{1}{n-1}} - (\frac{1}{n})^{\frac{n}{n-1}}$.*

- *For $f(x) = (1-\sqrt{x})^2$ we have the hellinger distance: $\int_0^1 \left(\sqrt{n}u^{\frac{n-1}{2}} - 1\right)^2 du = \int_0^1 (nu^{n-1} - 2\sqrt{n}u^{\frac{n-1}{2}} + 1)du = u^n - 2\sqrt{n}\frac{u^{\frac{n+1}{2}}}{\frac{n+1}{2}} + u\Big|_0^1 = 2(1 - \frac{2\sqrt{n}}{n+1}) = 2\frac{(1-\sqrt{n})^2}{n+1}$*

- *For $f(x) = -\log(x)$, we obtain the forward KL and we have : $\int_0^1 f(nu^{n-1})du = n - 1 - \log(n)$.*

**Guarantees with Rényi Divergence**  Turning now to the Rényi divergence for $\alpha \in (0,1) \cup (1,\infty)$:

$$D_\alpha(P||Q) = \frac{1}{(\alpha - 1)} \log \left( \int p^\alpha(x) q^{1-\alpha}(x) dx \right)$$

the limit as $\alpha \to 1$ $D_1(P||Q)) = \mathsf{KL}(P||Q)$ .

---

**Theorem 6** *Under Assumption 2 the best of n policy satisfies:*

$$D_\alpha(\pi_{r,\text{ref}}^{(n)}||\pi_{\text{ref}}) \leq \frac{1}{(\alpha - 1)} \log \left( \frac{n^\alpha}{\alpha(n-1)+1} \right) \tag{33}$$

---

**Proof 4 (Proof of Theorem 6)** *Applying DPI that holds also for the Rényi divergence twice from $Y, Y^{(n)}$ to $R, R^{(n)}$ and from $R, R^{(n)}$ to $E, E^{(n)}$ we obtain :*

$$D_\alpha(\pi_{r,\text{ref}}^{(n)}||\pi_{ref}|X) \leq D_\alpha(E^{(n)}||E)$$

$$D_\alpha(E^{(n)}||E) = \frac{1}{(\alpha - 1)} \log \left( \int_0^\infty n^\alpha (1 - e^{-x})^{\alpha(n-1)} e^{-\alpha x} e^{-x(1-\alpha)} dx \right)$$
$$= \frac{1}{(\alpha - 1)} \log \left( \int_0^{+\infty} n^\alpha (1 - e^{-x})^{\alpha(n-1)} e^{-x} dx \right)$$

*Let $u = 1 - e^{-x}$ we have $du = e^{-x} dx$*

$$D_\alpha(E^{(n)}||E) = \frac{1}{(\alpha - 1)} \log \left( \int_0^1 n^\alpha u^{\alpha(n-1)} du \right)$$
$$= \frac{1}{(\alpha - 1)} \left( \log n^\alpha + \log \int_0^1 u^{\alpha(n-1)} du \right)$$
$$= \frac{1}{(\alpha - 1)} \left( \log n^\alpha + \log \frac{u^{\alpha(n-1)+1}}{\alpha(n-1)+1}\Big|_0^1 \right)$$
$$= \frac{1}{(\alpha - 1)} \log \left( \frac{n^\alpha}{\alpha(n-1)+1} \right)$$

**From Renyi to KL guarantees**  Let $s_1(\alpha) = (\alpha - 1)$ , and $s_2(\alpha) = \log\left(\frac{n^\alpha}{\alpha(n-1)+1}\right)$, we have $D_\alpha(E^{(n)}||E) = \frac{s_2(\alpha)}{s_1(\alpha)}$ , we have $\mathsf{KL}(E^{(n)}||E) = \lim_{\alpha \to 1} D_\alpha(E^{(n)}||E) = \lim_{\alpha \to 1} \frac{s_2(\alpha)}{s_\alpha} = \frac{0}{0}$, hence applying L'Hôpital rule we have: $\lim_{\alpha \to 1} \frac{s_2(\alpha)}{s_1(\alpha)} = \lim_{\alpha \to 1} \frac{s_2'(\alpha)}{s_1'(\alpha)} = \lim_{\alpha \to 1} \frac{\log(n) - \frac{n-1}{\alpha(n-1)+1}}{1} = \log(n) - \frac{n-1}{n}$. Hence we recover the result for the $\mathsf{KL}$ divergence.

### B.3 Best of $n$ Dominance

**Proof 5 (Proof of Proposition 1 )** $F_{E^{(n)}}(x) = (F_E(x))^n \leq F_E(x), \forall x \geq 0$, *which means also that* $F_{E^{(n)}}^{-1}(t) \geq F_E^{-1}(t), \forall t \in [0,1]$, *which means that* $E^{(n)}$ *dominates* $E$ *in the first stochastic order :* $E^{(n)} \underset{FSD}{\succcurlyeq} E$ *, which means there exists a coupling between* $E^{(n)}$ *and* $E$, $\pi \in \Pi(E^{(n)}, E)$, *such that* $E \geq e$, *for all* $(E, e) \sim \pi$. *On the other hand By Rényi and Monge map representations we have:* $R^{(n)} = F_R^{-1} \circ F_E(E^{(n)})$ *and* $R = F_R^{-1} \circ F_E(E)$, *given that* $T = F_R^{-1} \circ F_E$ *is non decreasing the same coupling* $\pi$ *guarantees that* $T(E) \geq T(e)$, *for all* $(E, e) \sim \pi$ *and Hence* $R^{(n)} \underset{FSD}{\succcurlyeq} R$.

**Corollary 2** *Best of n-polciy has higher expectation :*

$$\mathbb{E}R^{(n)} \geq \mathbb{E}R,$$

*and is a safer policy, let the Tail Value at Risk be:*

$$\text{TVAR}_p(X) = \frac{1}{p}\int_0^p Q_R(t)dt$$

*We have*

$$\text{TVAR}_p(R^n) \geq \text{TVAR}_p(R), \forall p \in [0,1]$$

**Proof 6 (Proof of Corollary 2)** *First order dominance implies second order dominance (i.e by integrating quantiles). Expectation is obtained for* $p = 1$.

## C   Best of $n$ and RL Policy

**Proof 7 (Proof of Proposition 2)** *We fix here* $\gamma = \frac{1}{\lambda_\Delta}$

$$\mathsf{KL}(\pi_{r,\text{ref}}^{(n)}||\pi_{\gamma,r}) = \int \pi_{r,\text{ref}}^{(n)}(y|x)\log\left(\frac{\pi_{r,\text{ref}}^{(n)}(y|x)}{\pi_{\gamma,r}(y|x)}\right) = \int \pi_{r,\text{ref}}^{(n)}(y|x)\log\left(\frac{\pi_{r,\text{ref}}^{(n)}(y|x)}{\pi_{\text{ref}}(y|x)\frac{e^{\gamma r(x,y)}}{Z_\gamma(x)}}\right)$$

$$= \mathsf{KL}(\pi_{r,\text{ref}}^{(n)}||\pi_{\text{ref}}) + \log\left(\mathbb{E}_{\pi_{\text{ref}}}e^{\gamma r}\right) - \gamma\int rd\pi_{r,\text{ref}}^{(n)}$$

*On the other hand by optimality of* $\pi_{\gamma,r}$ *we have:*

$$\mathsf{KL}\left(\pi_{\gamma,r}||\pi_{\text{ref}}\right) = \gamma\int rd\pi_{\gamma,r} - \log\left(\int e^{\gamma r}d\pi_{\text{ref}}\right)$$

*and hence we have:*

$$\mathsf{KL}(\pi_{r,\text{ref}}^{(n)}||\pi_{\gamma,r}) = \mathsf{KL}(\pi_{r,\text{ref}}^{(n)}||\pi_{\text{ref}}) - \mathsf{KL}\left(\pi_{\gamma,r}||\pi_{\text{ref}}\right) + \gamma\left(\int rd\pi_{\gamma,r} - \int rd\pi_{r,\text{ref}}^{(n)}\right)$$

*We choose n such that :*

$$\mathsf{KL}(\pi_{r,\text{ref}}^{(n)}||\pi_{\text{ref}}) \leq \log(n) - \frac{n-1}{n} \leq \mathsf{KL}\left(\pi_{\gamma,r}||\pi_{\text{ref}}\right) = \Delta$$

*and we conclude choosing* $n = e^\Delta$ *therefore for that choice of n that:*

$$\mathsf{KL}(\pi_{r,\text{ref}}^{(n)}||\pi_{\gamma,r}) \leq \gamma\left(\int rd\pi_{\gamma,r} - \int rd\pi_{r,\text{ref}}^{(n)}\right)$$

*On the other hand we have:*

$$\left| \int r d\pi_{\gamma,r} - \int r d\pi_{r,\text{ref}}^{(n)} \right| = \left| \int r \exp(\gamma r) \frac{1}{Z_\gamma} d\pi_{\text{ref}} - \int \max_i r(x_i) d\pi_{\text{ref}}(x_1) \dots d\pi_{\text{ref}}(x_n) \right|$$

$$= \left| \int \left( \frac{1}{n} \sum_{i=1}^n \frac{r(x_i) \exp(\gamma r(x_i))}{Z_\gamma} - \max_i r(x_i) \right) d\pi_{\text{ref}}(x_1) \dots d\pi_{\text{ref}}(x_n) \right|$$

$$= \left| \int \left( \frac{1}{n} \sum_{i=1}^n \frac{r(x_i) \exp(\gamma r(x_i))}{\sum_{i=1}^n \exp(\gamma r(x_i))} \frac{\sum_{i=1}^n \exp(\gamma r(x_i))}{Z_\gamma} - \max_i r(x_i) \right) d\pi_{\text{ref}}(x_1) \dots d\pi_{\text{ref}}(x_n) \right|$$

$$\leq \int \left| \max r(x_i) \left( \frac{\frac{1}{n} \sum_{i=1}^n \exp(\gamma r(x_i))}{Z_\gamma} - 1 \right) \right| d\pi_{\text{ref}}(x_1) \dots d\pi_{\text{ref}}(x_n)$$

$$\leq \frac{M}{Z_\gamma} \mathbb{E} \left| \frac{1}{n} \sum_{i=1}^n \exp(\gamma r(x_i)) - Z_\gamma \right|$$

*where we used the following fact, followed by Jensen inequality :*

$$\sum_{i=1}^n \frac{r(x_i) \exp(\gamma r(x_i))}{\sum_{i=1}^n \exp(\gamma r(x_i))} \leq \max_i r(x_i).$$

*Assume that the reward is bounded hence we have by Hoeffding inequality :*

$$\mathbb{P} \left( \left| \frac{1}{n} \sum_{i=1}^n \exp(\gamma r(x_i)) - Z_\gamma \right| \geq t \right) \leq 2 e^{-\frac{nt^2}{2(\exp(\gamma M) - \exp(-\gamma M))^2}}$$

*Hence we have:*

$$\mathbb{E} \left| \frac{1}{n} \sum_{i=1}^n \exp(\gamma r(x_i)) - Z_\gamma \right| \leq 2 \sqrt{\frac{\pi}{2}} \frac{\exp(\gamma M) - \exp(-\gamma M)}{\sqrt{n}}$$

$$\mathsf{KL}(\pi_{r,\text{ref}}^{(\exp(\Delta))} || \pi_{\lambda_\Delta, r}) \leq \frac{M}{\lambda_\Delta Z_{1/\lambda_\Delta}} \sqrt{2\pi} (\exp(\gamma M) - \exp(-\gamma M)) \sqrt{\exp(-\Delta)}.$$

## D  Transportation Inequalities and KL Divergence

### D.1  Transportation Inequalities with KL

The following Lemma (Lemma 4.14 in (Boucheron et al., 2013)) uses the Donsker-Varadhan representation of the KL divergence to obtain bounds on the change of measure , and using the tails of $\pi_{\text{ref}}$.

**Lemma 4 (Lemma 4.14 in (Boucheron et al., 2013))** *Let $\psi$ be a convex and continuously differentiable function $\psi$ on a possibly unbounded interval $[0, b)$, and assume $\psi(0) = \psi'(0) = 0$. Define for every $x \geq 0$, the convex conjugate $\psi^*(x) = \sup_{\lambda \in [0,b)} \lambda x - \psi(\lambda)$ , and let $\psi^{*-1}(t) = \inf\{x \geq 0 : \psi^*(x) > t\}$. Then the following statements are equivalent:*
*(i) For $\lambda \in [0, b)$*

$$\log \left( \int e^{\lambda(r - \int r dQ)} dQ \right) \leq \psi(\lambda),$$

*(ii) For any probability measure $P$ that is absolutely continuous with respect to $Q$ and such that $\mathsf{KL}(P||Q) < \infty$:*

$$\int r dP - \int r dQ \leq \psi^{*-1}(\mathsf{KL}(P||Q)).$$

**Lemma 5 ( Inverse of the conjugate (Boucheron et al., 2013))**      *1. If $Q \in \mathsf{SubGauss}(\sigma^2)$, we have for $t \geq 0$ $\psi^{*-1}(t) = \sqrt{2\sigma^2 t}$.*

*2. If $Q \in \mathsf{Subgamma}(\sigma^2, c)$, we have for $t \geq 0$ $\psi^{*-1}(t) = \sqrt{2\sigma^2 t} + ct$.*

We give here a direct proof for the subgaussian case:

**Proof 8** *By the Donsker Varadhan representation of the* $\mathsf{KL}$ *we have:*

$$\mathsf{KL}(P||Q) = \sup_{h} \int h \, dP - \log\left(\int e^h \, dQ\right)$$

*Fix $x$ and $M > 0$ and define for $0 < \lambda < M$*

$$h_\lambda(y) = \lambda\left(r(x,y) - \mathbb{E}_{\pi_{\mathrm{ref}}(y|x)} r(x,y)\right)$$

*We omit in what follows $x$ and $y$, but the reader can assume from here on that $\pi$ and $\pi_{\mathrm{ref}}$ are conditioned on $x$. Note that $R_{\mathrm{ref}}|x = (r(x,.))_\sharp \pi_{\mathrm{ref}}(.|x)$ and we assume $R_{\mathrm{ref}}|x$ subgaussian. Note that*

$$\mathbb{E}_{\pi_{\mathrm{ref}}} e^{h_\lambda} = \mathbb{E}_{\pi_{\mathrm{ref}}|x} e^{\lambda(r - \mathbb{E}_{\pi_{\mathrm{ref}}|x} r)} = M_{R_{\mathrm{ref}}|x}(\lambda),$$

*where $M_{R_{\mathrm{ref}}|x}$ the moment generating function of the reward under the reference policy. $R_{\mathrm{ref}}|x$ is subgaussian we have for all $\lambda \in \mathbb{R}$:*

$$\mathbb{E}_{\pi_{\mathrm{ref}}|x} e^{h_\lambda} \leq e^{\frac{\lambda^2 \sigma^2}{2}} \leq e^{\frac{M^2 \sigma^2}{2}} < \infty$$

*Hence $h_\lambda \in \mathcal{H}$ and we have for all $\pi \ll \pi_{\mathrm{ref}}$ and for all $0 < M < \infty$ and $0 < \lambda < M$:*

$$\lambda \mathbb{E}_{\pi|x}(r - \mathbb{E}_{\pi_{\mathrm{ref}}|x} r) \leq \mathsf{KL}(\pi||\pi_{\mathrm{ref}}|x) + \log\left(\mathbb{E}_{\pi_{\mathrm{ref}}|x} e^{\lambda(r - \mathbb{E}_{\pi_{\mathrm{ref}}|x} r)}\right)$$

*or equivalently:*

$$\mathbb{E}_{\pi|x} r - \mathbb{E}_{\pi_{\mathrm{ref}}|x} r \leq \frac{1}{\lambda}\mathsf{KL}(\pi||\pi_{\mathrm{ref}}|x) + \frac{1}{\lambda}\log\left(\mathbb{E}_{\pi_{\mathrm{ref}}|x} e^{\lambda(r - \mathbb{E}_{\pi_{\mathrm{ref}}|x} r)}\right)$$

*Finally we have for $\pi \ll \pi_{\mathrm{ref}}$ for all $0 < \lambda < M$:*

$$\mathbb{E}_{\pi|x} r - \mathbb{E}_{\pi_{\mathrm{ref}}|x} r \leq \frac{1}{\lambda}\mathsf{KL}(\pi||\pi_{\mathrm{ref}}|x) + \frac{1}{\lambda}\log\left(M_{R_{\mathrm{ref}}|x}(\lambda)\right) \tag{34}$$

*Being a subgaussian, the MGF of $R_{\mathrm{ref}}|x$ is bounded as follows:*

$$\log\left(M_{R_{\mathrm{ref}}|x}(\lambda)\right) \leq \frac{\lambda^2 \sigma^2}{2}.$$

*Hence we have for :*

$$\mathbb{E}_{\pi|x} r - \mathbb{E}_{\pi_{\mathrm{ref}}|x} r \leq \frac{1}{\lambda}\mathsf{KL}(\pi||\pi_{\mathrm{ref}}|x) + \frac{\lambda \sigma^2}{2}$$

*Integrating over $x$ we obtain for all $\pi \ll \pi_{\mathrm{ref}}$ and all $0 < \lambda < M$:*

$$\mathbb{E}_\pi r - \mathbb{E}_{\pi_{\mathrm{ref}}} r \leq \frac{1}{\lambda}\mathsf{KL}(\pi||\pi_{\mathrm{ref}}) + \frac{\lambda \sigma^2}{2}$$

*Define :*

$$\delta(\lambda) = \frac{1}{\lambda}\mathsf{KL}(\pi||\pi_{\mathrm{ref}}) + \frac{\lambda \sigma^2}{2}$$

*minimizing the upper bound $\delta(\lambda)$ for $\lambda \in (0, M]$, taking derivative $\delta'(\lambda) = -\frac{\mathsf{KL}(\pi||\pi_{\mathrm{ref}})}{\lambda^2} + \frac{\sigma^2}{2} = 0$ gives $\lambda^* = \sqrt{\frac{2\mathsf{KL}(\pi||\pi_{\mathrm{ref}})}{\sigma^2}}$. Taking $M = 2\lambda^*$, $\lambda^*$ is the minimizer. Putting this in the bound we have finally for all rewards $r$ for all $\pi$:*

$$\mathbb{E}_\pi r - \mathbb{E}_{\pi_{\mathrm{ref}}} r \leq \sqrt{2\sigma^2 \mathsf{KL}(\pi||\pi_{\mathrm{ref}})}. \tag{35}$$

**Proof 9 (Proof of Corollary 1)** *(i) This follows from optimality of $\pi_{\lambda_\Delta}$ and applying the transportation inequality for gaussian tail.*

*(ii) This follows from applying Corollary 2 (best of n policy has larger mean ) and Theorem 1 for bounding the* KL.

**Proof 10 (Proof of Theorem 2)** *For the penalized RL we have by optimality:*

$$\int r d\pi_{\gamma,r} - \frac{1}{\gamma}\mathsf{KL}(\pi_{\gamma,r}||\pi_{\mathrm{ref}}) = \frac{1}{\gamma}\log\left(\int e^{\gamma r} d\pi_{\mathrm{ref}}\right)$$

$$= \frac{1}{\gamma}\log\left(\int e^{\gamma(r-\int r d\pi_{\mathrm{ref}})} d\pi_{\mathrm{ref}}\right) + \int r d\pi_{\mathrm{ref}}$$

*It follows that :*

$$\frac{1}{\gamma}\log\left(\int e^{\gamma(r-\int r d\pi_{\mathrm{ref}})} d\pi_{\mathrm{ref}}\right) = \int r d\pi_{\gamma,r} - \int r d\pi_{\mathrm{ref}} - \frac{1}{\gamma}\mathsf{KL}(\pi_{\gamma,r}||\pi_{\mathrm{ref}}) \tag{36}$$

*On the other hand by the variational representation of the Rényi divergence we have:*

$$\int r d\pi_{\gamma,r} - \int r d\pi_{\mathrm{ref}} \leq \frac{D_\gamma(\pi_{\gamma,r}||\pi_{\mathrm{ref}})}{\gamma} - \frac{1}{\gamma-1}\log\left(\int e^{(\gamma-1)(r-\int r d\pi_{\gamma,r})} d\pi_{\gamma,r}\right)$$

$$+ \frac{1}{\gamma}\log\left(\int e^{\gamma(r-\int r d\pi_{\mathrm{ref}})} d\pi_{\mathrm{ref}}\right) \tag{37}$$

*Summing Equations equation 36 and equation 37 we obtain a bound on the moment generating function at $\gamma$ of $r_\sharp \pi_{\gamma,r}$ (this is not a uniform bound , it holds only for $\gamma$):*

$$\frac{1}{\gamma-1}\log\left(\int e^{(\gamma-1)(r-\int r d\pi_{\gamma,r})} d\pi_{\gamma,r}\right) \leq \frac{D_\gamma(\pi_{\gamma,r}||\pi_{\mathrm{ref}}) - \mathsf{KL}(\pi_{\gamma,r}||\pi_{\mathrm{ref}})}{\gamma}. \tag{38}$$

*Let us assume $\gamma > 1$ we have therefore the following bound on the logarithmic moment generation function at $\gamma-1$*

$$\psi_{r_\sharp \pi_{\gamma,r}}(\gamma-1) \leq \frac{\gamma-1}{\gamma}\left(D_\gamma(\pi_{\gamma,r}||\pi_{\mathrm{ref}}) - \mathsf{KL}(\pi_{\gamma,r}||\pi_{\mathrm{ref}})\right)$$

*Let $R_{i,\gamma} = r_\sharp \pi_{\gamma,r}, i = 1 \ldots m$ , the reward evaluation of $m$ independent samples of $\pi_{\gamma,r}$ we have:*

$$\mathbb{P}\left\{\sum_{i=1}^m (R_{i,\gamma} - \int r d\pi_{\gamma,r}) > mt\right\} = \mathbb{P}(e^{\sum_{i=1}^m (\gamma-1)(R_{i,\gamma} - \int r d\pi_{\gamma,r})} > e^{m(\gamma-1)t})$$

$$\leq e^{-(\gamma-1)mt} e^{m\psi_{R_\gamma}(\gamma-1)}$$

$$\leq e^{-(\gamma-1)mt} e^{m\frac{\gamma-1}{\gamma}(D_\gamma(\pi_{\gamma,r}||\pi_{\mathrm{ref}}) - \mathsf{KL}(\pi_{\gamma,r}||\pi_{\mathrm{ref}}))}$$

$$\leq e^{-m(\gamma-1)\left(t - \frac{D_\gamma(\pi_{\gamma,r}||\pi_{\mathrm{ref}}) - \mathsf{KL}(\pi_{\gamma,r}||\pi_{\mathrm{ref}})}{\gamma}\right)} \tag{39}$$

*Let $t_0 > 0$, hence we have for $\gamma > 1$:*

$$\mathbb{P}\left\{\frac{1}{m}\sum_{i=1}^m R_{i,\gamma} > \int r d\pi_{\gamma,r} + t_0 + \frac{D_\gamma(\pi_{\gamma,r}||\pi_{\mathrm{ref}}) - \mathsf{KL}(\pi_{\gamma,r}||\pi_{\mathrm{ref}})}{\gamma}\right\} \leq e^{-m(\gamma-1)t_0}$$

*Now turning to $R_{\mathrm{ref}} = r_\sharp \pi_{\mathrm{ref}}$, since $R_{\mathrm{ref}} \in \mathsf{SubGauss}(\sigma_{\mathrm{ref}}^2)$ we have for every $t_0 > 0$ :*

$$\mathbb{P}\left\{-\frac{1}{m}\sum_{i=1}^m R_{i,\mathrm{ref}} > -\int r d\pi_{\mathrm{ref}} + t_0\right\} \leq e^{-\frac{mt_0^2}{2\sigma_{\mathrm{ref}}^2}}$$

*Hence we have with probability at least $1 - e^{-\frac{m t_0^2}{2\sigma_{\mathrm{ref}}^2}} - e^{-m(\gamma-1)t_0}$ :*

$$\frac{1}{m}\sum_{i=1}^{m} R_{i,\gamma} - \frac{1}{m}\sum_{i=1}^{m} R_{i,\mathrm{ref}} \leq \int r d\pi_{\gamma,r} - \int r d\pi_{\mathrm{ref}} + 2t_0 + \frac{D_\gamma(\pi_{\gamma,r}||\pi_{\mathrm{ref}}) - \mathsf{KL}(\pi_{\gamma,r}||\pi_{\mathrm{ref}})}{\gamma}$$

$$\leq \sqrt{2\sigma_{\mathrm{ref}}^2 \mathsf{KL}(\pi||\pi_{\mathrm{ref}})} + 2t_0 + \frac{D_\gamma(\pi_{\gamma,r}||\pi_{\mathrm{ref}}) - \mathsf{KL}(\pi_{\gamma,r}||\pi_{\mathrm{ref}})}{\gamma}.$$

# E  Proofs for Transportation Inequalities and Rényi Divergence

**Proposition 6 (Fenchel Conjugate Propreties)** *Let $F$ and $G$ be convex functions on a space $E$ and $F^*$, $G^*$ be their convex conjugates defined on $E^*$. We have:*

*1. Let $F_\gamma(x) = \gamma F(x)$ we have:*

$$F_\gamma^*(p) = \gamma F^*\left(\frac{p}{\gamma}\right) \tag{40}$$

*2. Duality:*

$$\min_{x \in E} F(x) + G(x) = \max_{p \in E^*} -F^*(-p) - G^*(p) \tag{41}$$

*3. Toland Duality:*

$$\min_{x \in E} F(x) - G(x) = \min_{p} G^*(p) - F^*(p) \tag{42}$$

**Proof 11 (Proof of Theorem 3)** *Let $\gamma > 0$, let $F_{P,\gamma}(R) = \gamma \mathsf{KL}(R||P)$, the Fenchel conjugate of $F_{P,1}(.)$ is defined for $h$ bounded and measurable function as follows $F_{P,1}^*(h) = \log \mathbb{E}_P e^h$. It follows by 1) in Proposition 6 that : $F_{P,\gamma}^*(h) = \gamma F_{P,1}^*(\frac{h}{\gamma}) = \gamma \log \mathbb{E}_P e^{\frac{h}{\gamma}}$.*
*For $0 < \alpha < 1$: The objective function in equation 15 is the sum of convex functions: $F_{P,\alpha}(R) + F_{Q,1-\alpha}(R)$, by (2) in Proposition 6, we have by duality:*

$$(1-\alpha)D_\alpha(P||Q) = \inf_R F_{P,\alpha}(R) + F_{Q,1-\alpha}(R)$$

$$= \sup_{h \in \mathcal{H}} -F_{P,\alpha}^*(-h) - F_{Q,1-\alpha}^*(h)$$

$$= \sup_{h \in \mathcal{H}} -\alpha \log \mathbb{E}_P e^{-\frac{h}{\alpha}} - (1-\alpha) \log \mathbb{E}_Q e^{\frac{h}{1-\alpha}}$$

*Replacing $h$ by $(1-\alpha)(\alpha)h$ does not change the value of the sup and hence we obtain:*

$$(1-\alpha)D_\alpha(P||Q) = \sup_{h \in \mathcal{H}} -\alpha \log \mathbb{E}_P e^{-\frac{(1-\alpha)(\alpha)h}{\alpha}} - (1-\alpha) \log \mathbb{E}_Q e^{\frac{(1-\alpha)(\alpha)h}{1-\alpha}}$$

$$= \sup_{h \in \mathcal{H}} -\alpha \log \mathbb{E}_P e^{-(1-\alpha)h} - (1-\alpha) \log \mathbb{E}_Q e^{\alpha h}.$$

*dividing by $\frac{1}{\alpha(1-\alpha)}$ both sides we obtain for $0 < \alpha < 1$:*

$$\frac{1}{\alpha}D_\alpha(P||Q) = \sup_{h \in \mathcal{H}} -\frac{1}{1-\alpha} \log \mathbb{E}_P e^{-(1-\alpha)h} - \frac{1}{\alpha} \log \mathbb{E}_Q e^{\alpha h}$$

*For $\alpha > 1$: The objective function in equation 15 is the difference of convex functions: $F_{P,\alpha}(R) - F_{Q,\alpha-1}(R)$, by Toland Duality (3) in Proposition 6 we have:*

$$(1-\alpha)D_\alpha(P||Q) = \inf_R F_{P,\alpha}(R) - F_{Q,\alpha-1}(R)$$

$$= \inf_{h \in \mathcal{H}} F_{Q,\alpha-1}^*(h) - F_{P,\alpha}^*(h)$$

$$= \inf_{h \in \mathcal{H}} (\alpha-1) \log \mathbb{E}_Q e^{\frac{h}{(\alpha-1)}} - \alpha \log \mathbb{E}_P e^{\frac{h}{\alpha}}$$

*The inf does not change when we replace $h$ by $\alpha(\alpha-1)h$, hence we have:*

$$(\alpha-1)D_\alpha(P||Q) = -\inf_{h\in\mathcal{H}}(\alpha-1)\log\mathbb{E}_Q e^{\frac{\alpha(\alpha-1)h}{(\alpha-1)}} - \alpha\log\mathbb{E}_P e^{\frac{\alpha(\alpha-1)h}{\alpha}}$$

$$= \sup_{h\in\mathcal{H}}\alpha\log\mathbb{E}_P e^{(\alpha-1)h} - (\alpha-1)\log\mathbb{E}_Q e^{\alpha h}$$

*dividing both sides by $\frac{1}{\alpha(\alpha-1)}$ we obtain for $\alpha > 1$:*

$$\frac{1}{\alpha}D_\alpha(P||Q) = \sup_{h\in\mathcal{H}}\frac{1}{\alpha-1}\log\mathbb{E}_P e^{(\alpha-1)h} - \frac{1}{\alpha}\log\mathbb{E}_Q e^{\alpha h}.$$

**Proof 12 (Proof of Lemma 2 )** *Adding and subtracting in the exponential $\int hdP$ and $\int hdQ$ resp we obtain the result:* $\frac{1}{\alpha-1}\log\left(\int e^{(\alpha-1)h}dP\right) - \frac{1}{\alpha}\log\left(\int e^{\alpha h}dQ\right) = \frac{1}{\alpha-1}\log\left(\int e^{(\alpha-1)(h-\int hdP+\int hdP)}dP\right) - \frac{1}{\alpha}\log\left(\int e^{\alpha(h-\int hdQ+\int hdQ)}dQ\right) = \int hdP - \int hdQ + \frac{1}{\alpha-1}\log\left(\int e^{(\alpha-1)(h-\int hdP)}dP\right) - \frac{1}{\alpha}\log\left(\int e^{\alpha(h-\int hdQ)}dQ\right)$

**Proof 13 ( Proof of Lemma 3)** *Note that we have for $0 < \alpha < 1$, $\frac{1}{\alpha}D_\alpha(P||Q) = \frac{1}{1-\alpha}D_{1-\alpha}(Q||P)$ (See Proposition 2 in van Erven & Harremos (2014)). Taking limits we obtain $\lim_{\alpha\to 0}\frac{1}{\alpha}D_\alpha(P||Q) = D_1(Q||P) = \mathsf{KL}(Q||P)$.*

**Proof 14 (Proof of Theorem 4 )** *For $0 < \alpha < 1$, we have for all $h \in \mathcal{H}$ :*

$$\int hdP - \int hdQ \le \frac{1}{\alpha}D_\alpha(P||Q) + \frac{1}{1-\alpha}\log\left(\int e^{(\alpha-1)(h-\int hdP)}dP\right) + \frac{1}{\alpha}\log\left(\int e^{\alpha(h-\int hdQ)}dQ\right) \quad (43)$$

*Assuming $r$ is bounded $0 < r < b$ then we have $(r)_\sharp P - \mathbb{E}_P r$ and $(r)_\sharp Q - \mathbb{E}_Q r$ are sub-Gaussian with parameter $\sigma^2 = \frac{b^2}{4}$. Hence we have for $\lambda \in \mathbb{R}$:*

$$\mathbb{E}_P e^{\lambda(r-\int rdP)} \le \exp\left(\frac{\lambda^2\sigma_P^2}{2}\right) \;\; and \;\; \mathbb{E}_Q e^{\lambda(r-\int rdQ)} \le \exp\left(\frac{\lambda^2\sigma_Q^2}{2}\right),$$

*Fix a finite $M > 0$. For $0 < \lambda < M$ and $P = \pi|x$ and $Q = \pi_{\mathrm{ref}}|x$, consider $h_\lambda = \lambda r$, thanks to subgaussianity and boundedness of $\lambda$, $h_\lambda \in \mathcal{H}$ for all $\lambda \in (0, M)$. Hence we have by Equation equation 43 for all $\lambda \in (0, M)$:*

$$\lambda\left(\int rdP - \int rdQ\right) \le \frac{1}{\alpha}D_\alpha(P||Q) + \frac{1}{1-\alpha}\log\left(\int e^{\lambda(\alpha-1)(r-\int rdP)}dP\right) + \frac{1}{\alpha}\log\left(\int e^{\lambda\alpha(r-\int rdQ)}dQ\right)$$

*we have by sub-Gaussianity:*

$$\frac{1}{1-\alpha}\log\left(\int e^{\lambda(\alpha-1)(r-\int rdP)}dP\right) \le \frac{1}{1-\alpha}\frac{\lambda^2(1-\alpha)^2\sigma_P^2}{2} = \frac{\lambda^2(1-\alpha)\sigma_P^2}{2}$$

$$\frac{1}{\alpha}\log\left(\int e^{\lambda\alpha(r-\int rdQ)}dQ\right) \le \frac{1}{\alpha}\frac{\lambda^2\alpha^2\sigma_Q^2}{2} = \frac{\lambda^2\alpha\sigma_Q^2}{2}$$

*It follows that for all $\lambda \in (0, M)$*

$$\lambda\left(\int rd\pi|x - \int rd\pi_{\mathrm{ref}}|x\right) \le \frac{1}{\alpha}D_\alpha(\pi|x||\pi_{\mathrm{ref}}|x) + \frac{\lambda^2(1-\alpha)\sigma_P^2}{2} + \frac{\lambda^2\alpha\sigma_Q^2}{2}$$

$$= \frac{1}{\alpha}D_\alpha(\pi|x||\pi_{\mathrm{ref}}|x) + \frac{\lambda^2((1-\alpha)\sigma_P^2 + \alpha\sigma_Q^2)}{2}$$

*Integrating over x we obtain:*

$$\lambda \left( \int r d\pi - \int r d\pi_{\text{ref}} \right) \leq \frac{1}{\alpha} D_\alpha(\pi || \pi_{\text{ref}}) + \frac{\lambda^2((1-\alpha)\sigma_P^2 + \alpha\sigma_Q^2)}{2}$$

*Finally we have:*

$$\int r d\pi - \int r d\pi_{\text{ref}} \leq \frac{1}{\lambda\alpha} D_\alpha(\pi || \pi_{\text{ref}}) + \frac{\lambda((1-\alpha)\sigma_P^2 + \alpha\sigma_Q^2)}{2}$$

*minimizing over $\lambda \in (0, M)$: we obtain $\lambda^* = \sqrt{\frac{2D_\alpha(\pi || \pi_{\text{ref}})}{((1-\alpha)\sigma_P^2 + \alpha\sigma_Q^2)\alpha}}$, $M$ is free of choice, choosing $M = 2\lambda^*$, gives that $\lambda^*$ is the minimizer and hence we have for all $\alpha \in (0, 1)$:*

$$\int r d\pi - \int r d\pi_{\text{ref}} \leq \sqrt{\frac{2((1-\alpha)\sigma_P^2 + \alpha\sigma_Q^2)D_\alpha(\pi || \pi_{\text{ref}})}{\alpha}}.$$

## F   Goodhart Laws

**Proof 15 (Proof of Proposition 4)**  *We have by duality:*

$$\frac{1}{\gamma} \log \left( \int e^{\gamma r^*} d\pi_{\text{ref}} \right) = \sup_\nu \int r^* d\nu - \frac{1}{\gamma} \mathsf{KL}(\nu || \pi_{\text{ref}})$$

*hence for $\nu = \pi_{\gamma,r}$ we have:*

$$\frac{1}{\gamma} \log \left( \int e^{\gamma r^*} d\pi_{\text{ref}} \right) \geq \int r^* d\pi_{\gamma,r} - \frac{1}{\gamma} \mathsf{KL}(\pi_{\gamma,r} || \pi_{\text{ref}})$$

*Hence:*

$$\int r^* d\pi_{\gamma,r} \leq \frac{1}{\gamma} \log \left( \int e^{\gamma r^*} d\pi_{\text{ref}} \right) + \frac{1}{\gamma} \mathsf{KL}(\pi_{\gamma,r} || \pi_{\text{ref}})$$

*On the other hand by optimality of $\pi_{\gamma,r}$ we have:*

$$\mathsf{KL}\left( \pi_{\gamma,r} || \pi_{\text{ref}} \right) = \gamma \int r d\pi_{\gamma,r} - \log \left( \int e^{\gamma r} d\pi_{\text{ref}} \right)$$

*Hence we have:*

$$\int r^* d\pi_{\gamma,r} \leq \frac{1}{\gamma} \log \left( \int e^{\gamma r^*} d\pi_{\text{ref}} \right) + \int r d\pi_{\gamma,r} - \frac{1}{\gamma} \log \left( \int e^{\gamma r} d\pi_{\text{ref}} \right) \leq \int r d\pi_{\gamma,r} + \frac{1}{\gamma} \log \left( \frac{\int e^{\gamma r^*} d\pi_{\text{ref}}}{\int e^{\gamma r} d\pi_{\text{ref}}} \right)$$

*It follows that:*

$$\int r^* d\pi_{\gamma,r} - \int r^* d\pi_{\text{ref}} \leq \int r d\pi_{\gamma,r} - \int r d\pi_{\text{ref}} + \frac{1}{\gamma} \log \left( \frac{\int e^{\gamma(r^* - \int r^* d\pi_{\text{ref}})} d\pi_{\text{ref}}}{\int e^{\gamma(r - \int r d\pi_{\text{ref}})} d\pi_{\text{ref}}} \right)$$

$$\frac{\int e^{\gamma(r^* - \int r^* d\pi_{\text{ref}})} d\pi_{\text{ref}}}{\int e^{\gamma(r - \int r d\pi_{\text{ref}})} d\pi_{\text{ref}}} = \int e^{\gamma(r^* - r - (\int r^* d\pi_{\text{ref}} - \int r d\pi_{\text{ref}}))} \frac{e^{\gamma r} d\pi_{\text{ref}}}{\int e^{\gamma r} d\pi_{\text{ref}}}$$

$$= \int e^{\gamma(r^* - r - (\int r^* d\pi_{\text{ref}} - \int r d\pi_{\text{ref}}))} d\pi_{\gamma,r}$$

*Hence we have finally:*

$$\int r^* d\pi_{\gamma,r} - \int r^* d\pi_{\text{ref}} \leq \int r d\pi_{\gamma,r} - \int r d\pi_{\text{ref}} + \frac{1}{\gamma} \log \left( \int e^{\gamma(r^* - r - (\int r^* d\pi_{\text{ref}} - \int r d\pi_{\text{ref}}))} d\pi_{\gamma,r} \right)$$

$$\int r^* d\pi_{\gamma,r} - \int r^* d\pi_{\text{ref}} \leq \int r d\pi_{\gamma,r} - \int r d\pi_{\text{ref}} - \frac{1}{\gamma} \log \left( \int e^{\gamma(r - r^* - (\int r d\pi_{\text{ref}} - \int r^* d\pi_{\text{ref}}))} d\pi_{\gamma,r^*} \right)$$

*The proof follows from using the subgaussianity of $r_\sharp \pi_{\text{ref}}$ and the assumption on the soft max.*

**Proof 16 (Proof of Proposition 5)**

$$\mathbb{E}_\pi(r^* - r) - \mathbb{E}_{\pi_{\text{ref}}}(r^* - r) \leq 2||r - r^*||_\infty \mathsf{TV}(\pi, \pi_{\text{ref}})$$

*For $\pi_{r,\text{ref}}^{(n)}$, we have:*

$$\mathbb{E}_{\pi_{r,\text{ref}}^{(n)}}(r^*) - \mathbb{E}_{\pi_{\text{ref}}}(r^*) \leq \mathbb{E}_{\pi_{r,\text{ref}}^{(n)}}(r) - \mathbb{E}_{\pi_{\text{ref}}}(r) + 2||r - r^*||_\infty \mathsf{TV}(\pi_{r,\text{ref}}^{(n)}, \pi_{\text{ref}})$$

*and*

$$\mathbb{E}_{\pi_{r,\text{ref}}^{(n)}}(r^*) - \mathbb{E}_{\pi_{\text{ref}}}(r^*) \geq \mathbb{E}_{\pi_{r,\text{ref}}^{(n)}}(r) - \mathbb{E}_{\pi_{\text{ref}}}(r) - 2||r - r^*||_\infty \mathsf{TV}(\pi_{r,\text{ref}}^{(n)}, \pi_{\text{ref}})$$

*By the data processing inequality we have:* $\mathsf{TV}(\pi_{r,\text{ref}}^{(n)}, \pi_{\text{ref}}) \leq \mathsf{TV}(R_{r,\text{ref}}^{(n)}, R) = (\frac{1}{n})^{\frac{1}{n-1}} - (\frac{1}{n})^{\frac{n}{n-1}}$ *If $r$ has subguassian tails under $\pi_{\text{ref}}$ than we have:*

$$\mathbb{E}_{\pi_{r,\text{ref}}^{(n)}}(r^*) - \mathbb{E}_{\pi_{\text{ref}}}(r^*) \leq \sqrt{2\sigma^2 \left( \log(n) - \frac{n-1}{n} \right)} + 2||r - r^*||_\infty \left( (\frac{1}{n})^{\frac{1}{n-1}} - (\frac{1}{n})^{\frac{n}{n-1}} \right)$$

$$\mathbb{E}_{\pi_{r,\text{ref}}^{(n)}}(r^*) - \mathbb{E}_{\pi_{\text{ref}}}(r^*) \leq \sqrt{2\sigma^2 \left( \log(n) - \frac{n-1}{n} \right)} + 2 \inf_{r \in \mathcal{H}} ||r - r^*||_\infty \left( (\frac{1}{n})^{\frac{1}{n-1}} - (\frac{1}{n})^{\frac{n}{n-1}} \right).$$

# G   Supplementary Figures and Experiments

## G.1   Tails of Reward model FsfairX-LLaMA3-RM-v0.1 evaluated on Popular LLMs

Original_removing_missing_values_meta-llama_llama-2-13b-chat_p_0.9_temp_0.95_FSfair Distribution Plots

Figure 10: Reward evaluated on LLama2-7B. We see that the reward follows more a Gaussian or a gamma random variable and it is not heavy tailed. The Moment generating function (MGF) follows a quadratic The Hill index is not meaningful in this case.

## G.2   Reward versus KL on other LLMs with best of n policies

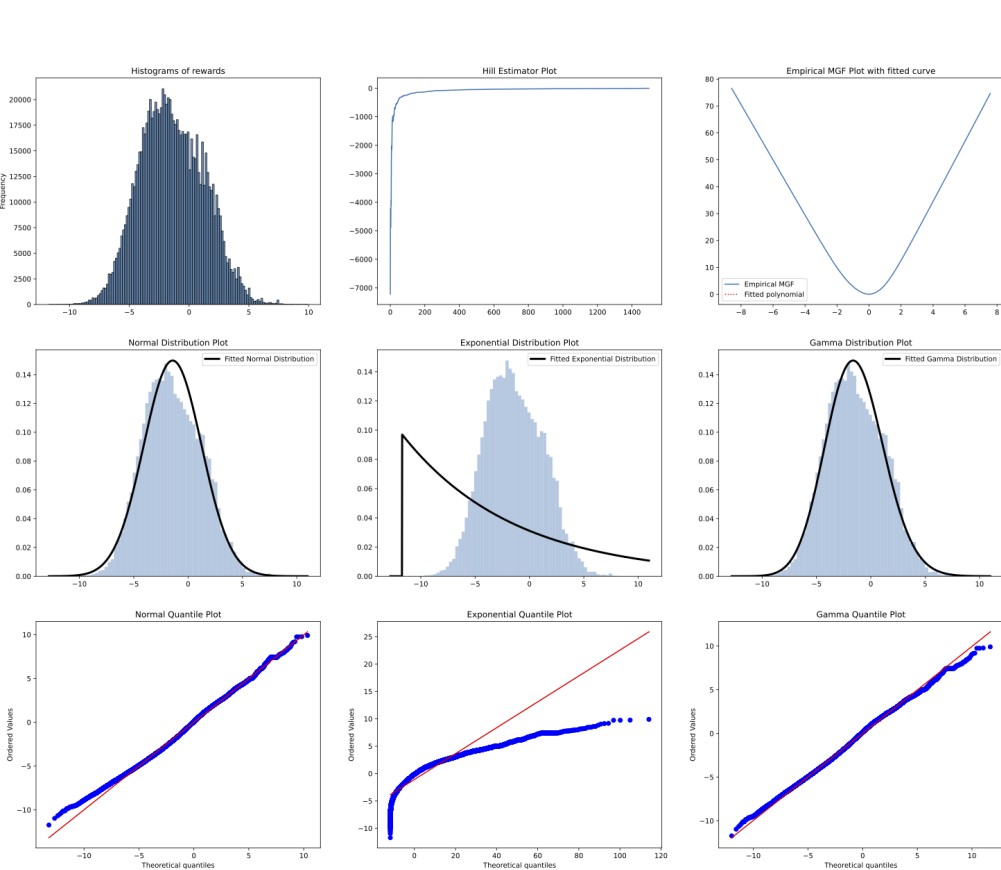

Figure 11: Reward evaluated on Merlinite 7B. The reward follows a gamma distribution as it is almostly perfectly matching it in the q-q plots (quantile plots).

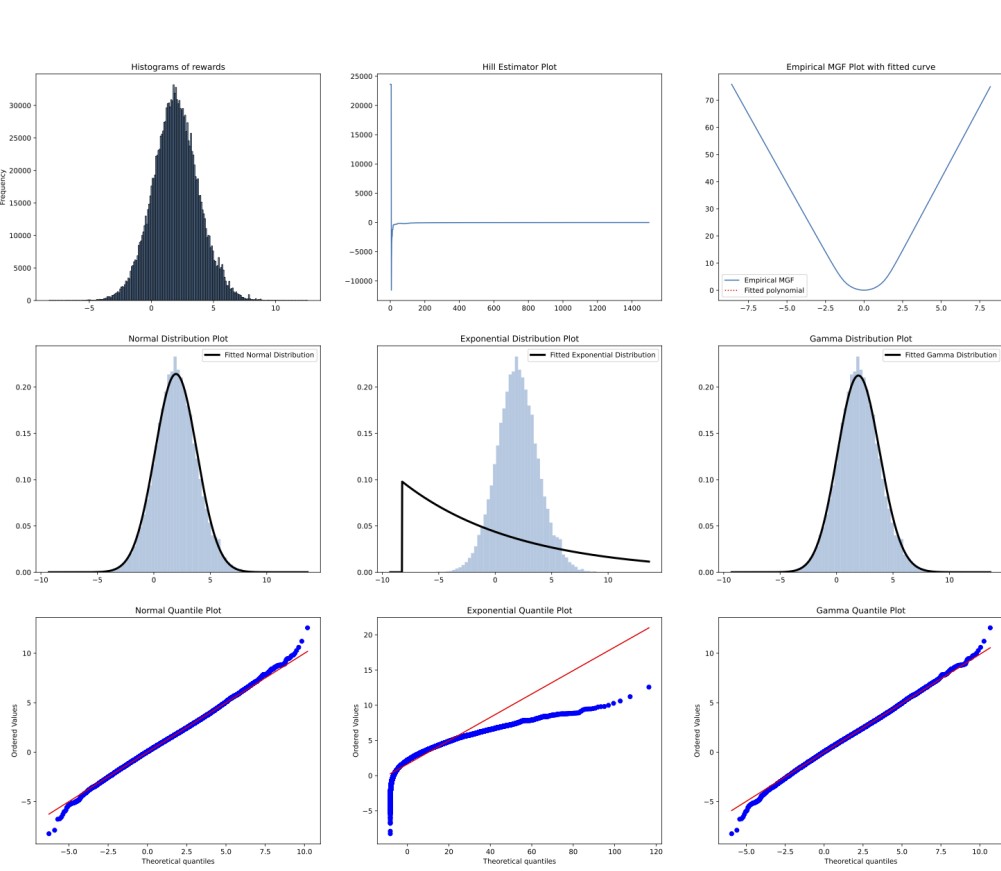

Figure 12: Reward evaluated on Mixtral8x7b. The reward follows a gaussian or a gamma distribution and is not heavy tailed.

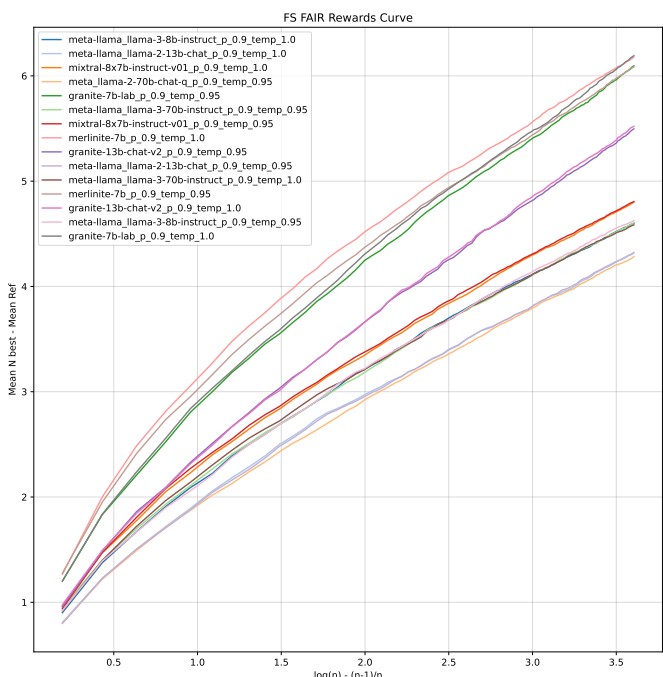

Figure 13: Centered Reward (FSFAIRX-LLAMA3-RM-V0.1) versus KL of best of n policy for various LLM

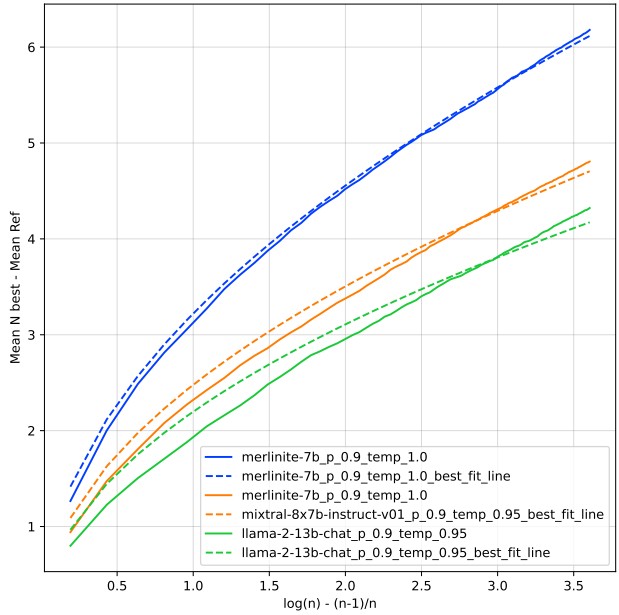

Figure 14: Best fit for $y = a * \sqrt{x}$ for centered best of n reward versus KL. We see that this fit is not as good as $y = a * \sqrt{x} + b * x$, hinting to a subgamma tail rather than subgaussian.

## H Experiments with OAS Reward Model OpenAssistant/reward-model-deberta-v3-large-v2

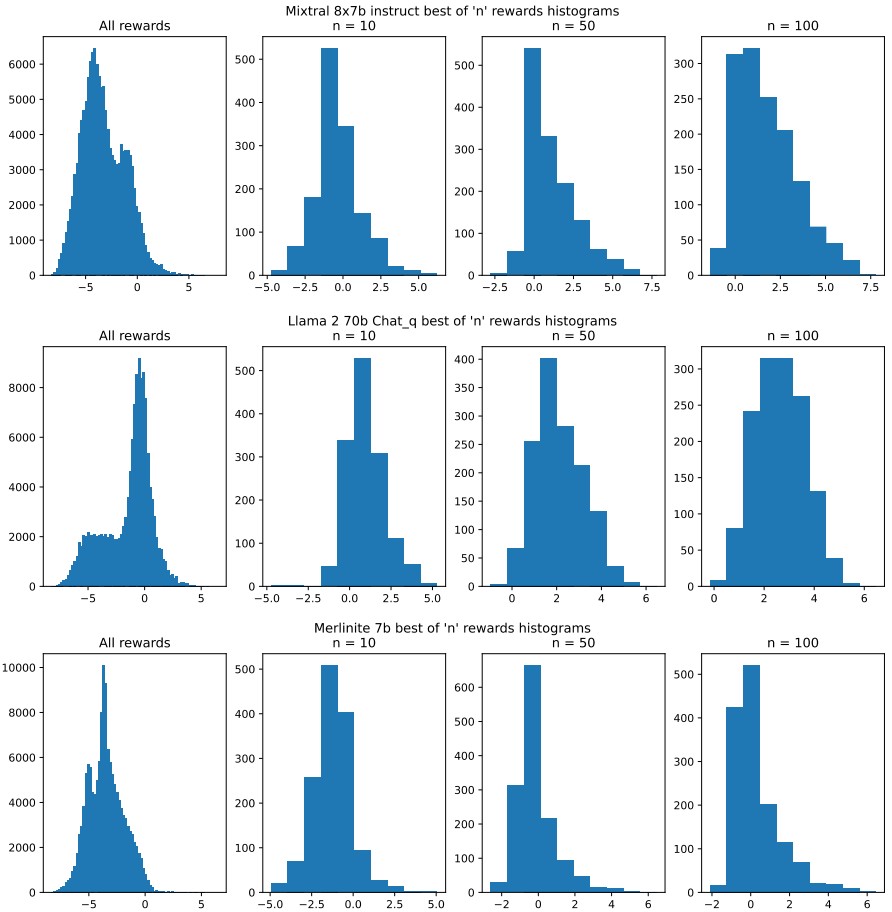

Figure 15: Histograms of OAS Reward for reference and best of n policies.

| Model and n | Mean | Std | Max |
|---|---|---|---|
| **Mixtral all** | -3.25 | 2.11 | 7.8 |
| **Mixtral n =10** | -0.34 | 1.5 | 6.23 |
| **Mixtral n =50** | 1.1 | 1.57 | 7.8 |
| **Mixtral n =100** | 1.76 | 1.64 | 7.8 |
| **Llama2_70b_chat all** | -1.47 | 2.17 | 6.48 |
| **Llama2_70b n =10** | 0.95 | 1.15 | 5.29 |
| **Llama2_70b n =50** | 2.12 | 1.09 | 6.48 |
| **Llama2_70b n =100** | 2.63 | 1.01 | 6.48 |
| **Merlinite all** | -3.47 | 1.6 | 6.46 |
| **Merlinite n =10** | -1.23 | 1.19 | 5.07 |
| **Merlinite n =50** | -0.14 | 1.08 | 6.46 |
| **Merlinite n =100** | 0.31 | 1.2 | 6.46 |

Table 4: Statistics of OPENASSISTANT/REWARD-MODEL-DEBERTA-V3-LARGE-V2 reward evaluated for reference and best of n policies for different $n$ values.

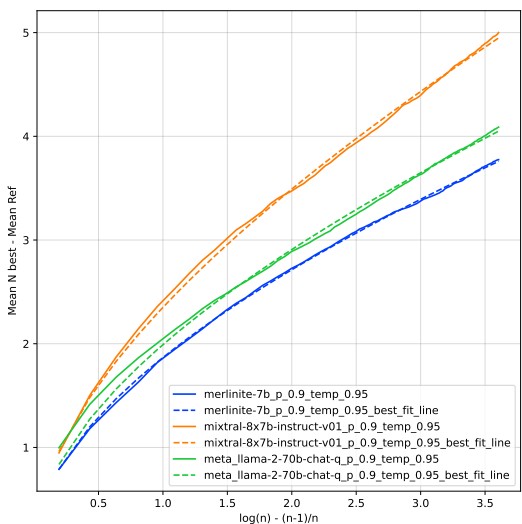

Figure 16: Centered Reward (OAS) versus KL best of n policies, with best fit $y = a\sqrt{x} + bx$. The fit hints to subgamma tails.

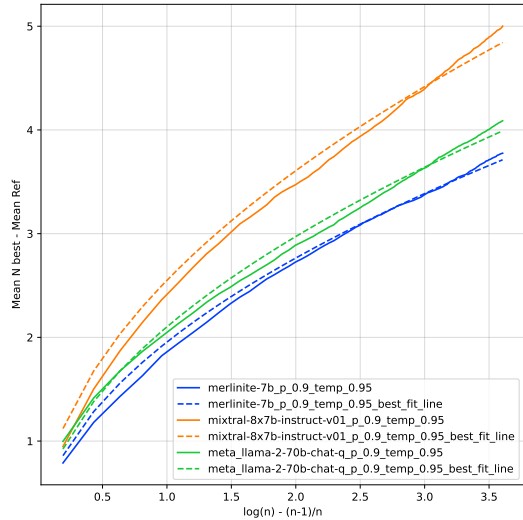

Figure 17: Centered Reward (OAS) versus KL best of n policies, with best fit $y = a\sqrt{x}$. This subgaussian fit provides an upper bound that is not as tight as above.

| Model | a | b |
|---|---|---|
| **Merlinite-7B** | 1.74119974 | 0.12555422 |
| **Mixtral-8x7B** | 2.06674977 | 0.28401807 |
| **Llama2-70B chat** | 1.83327634 | 0.15764552 |

Table 5: OAS Reward model: $y = a\sqrt{x} + bx$ best fitted coefficients for centered reward versus KL. $a$ is slightly larger than std.

| Model | a |
|---|---|
| **Merlinite-7B** | 1.9550695 |
| **Mixtral-8x7B** | 2.55054775 |
| **Llama-70B-Chat** | 2.10181062 |

Table 6: OAS Reward model: $y = a\sqrt{x}$ best fitted coefficients for centered reward versus KL.

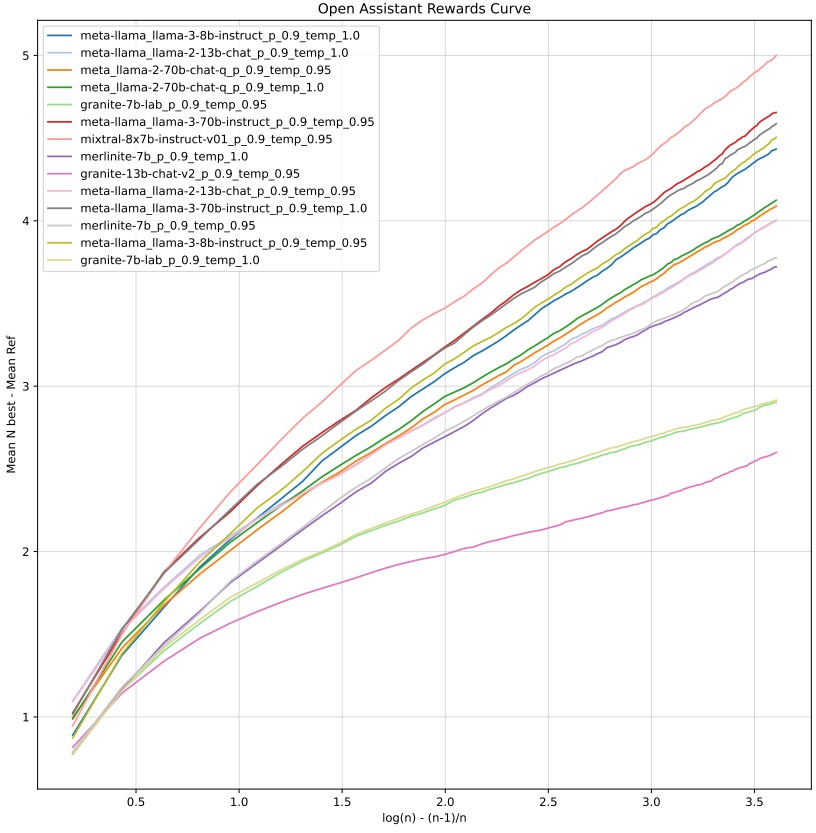

Figure 18: OAS Reward: Centered Rewards versus KL best of n policies for various models.

# I  Reward Versus Rényi In Best of N

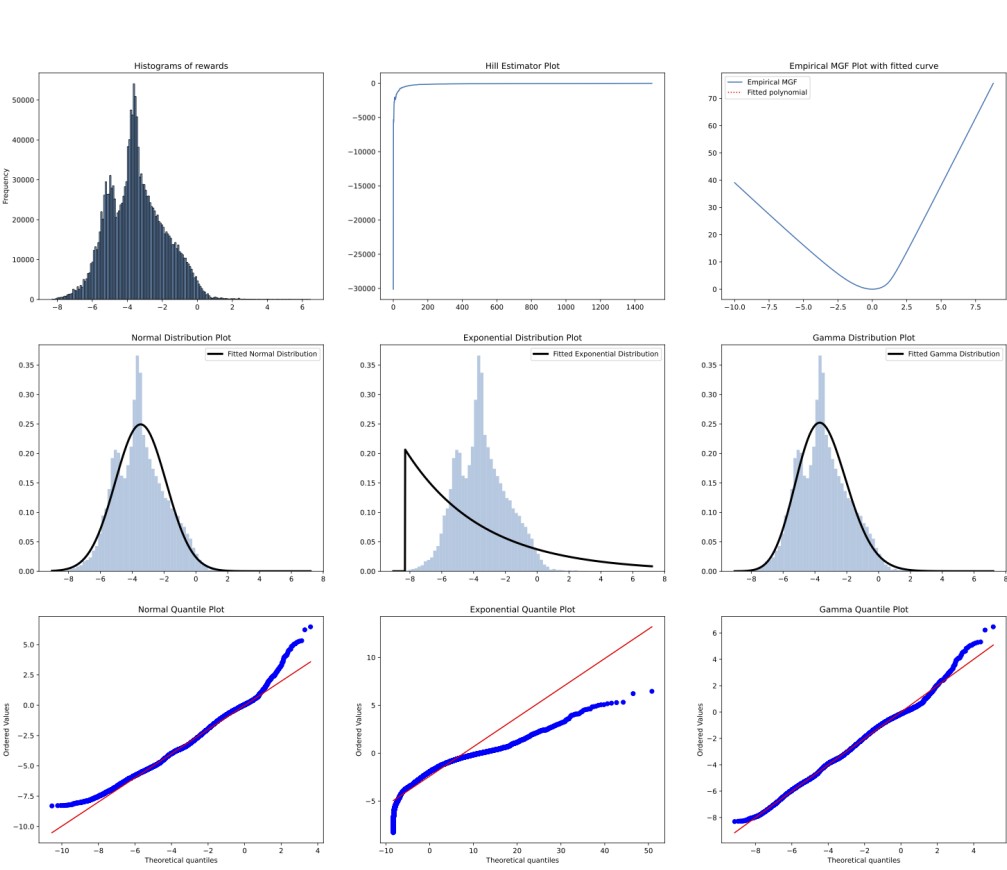

Figure 19: subgamma tails of the OAS reward for Merlinite 7B as seen in the q-q plots.

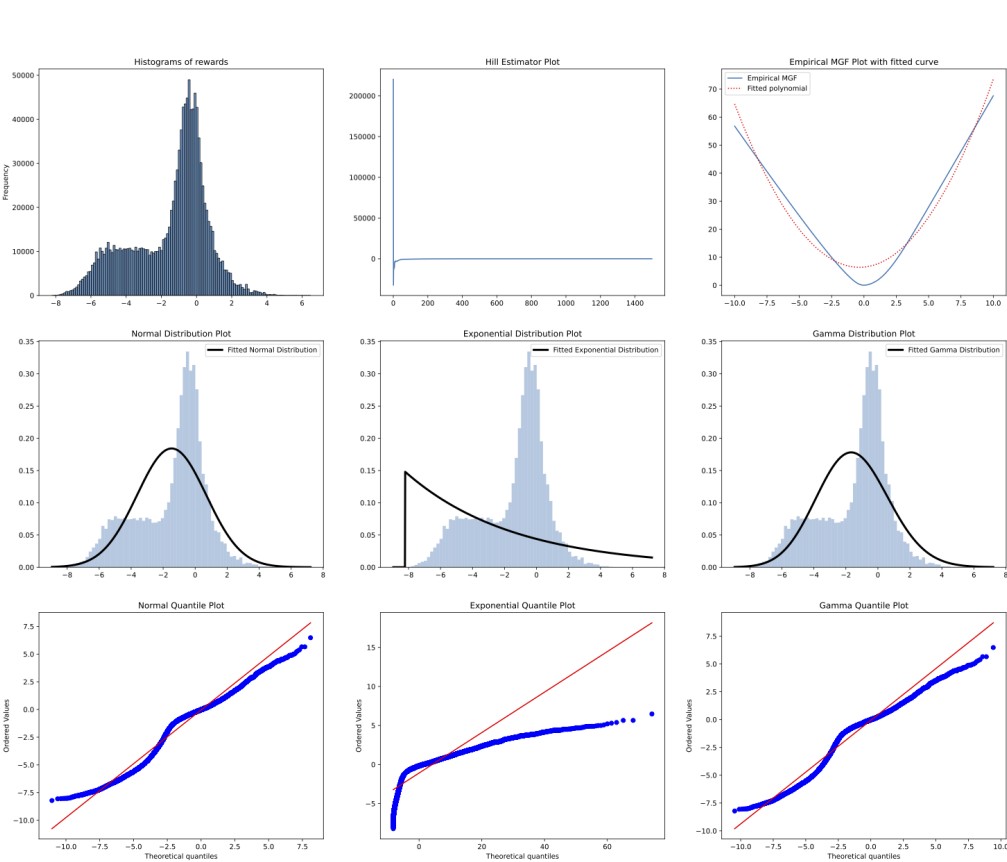

Figure 20: subgaussian/subgamma tails of the OAS reward for LLama2-70B-chat.

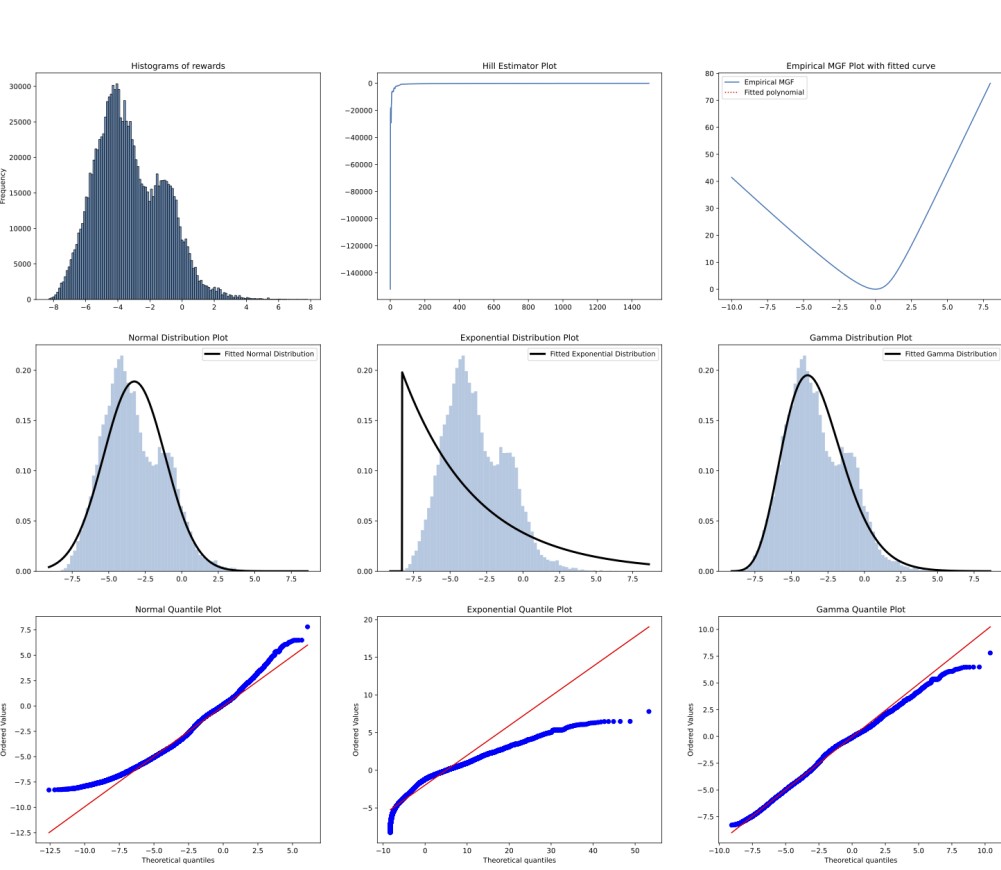

Figure 21: Subgamma tails of the OAS reward of Mixtral-8x7b-instruct.

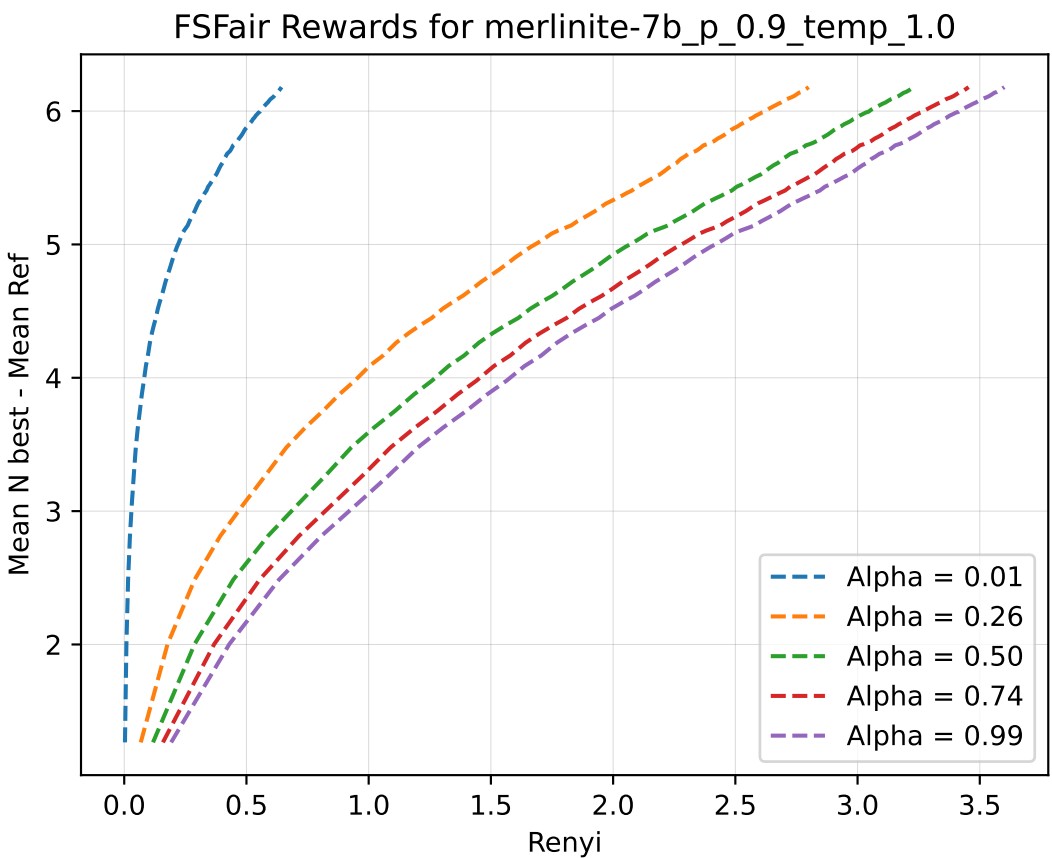

Figure 22: Centered Reward for Best of $N$ versus Renyi divergence for $\alpha \in (0, 1)$ - Merlinite

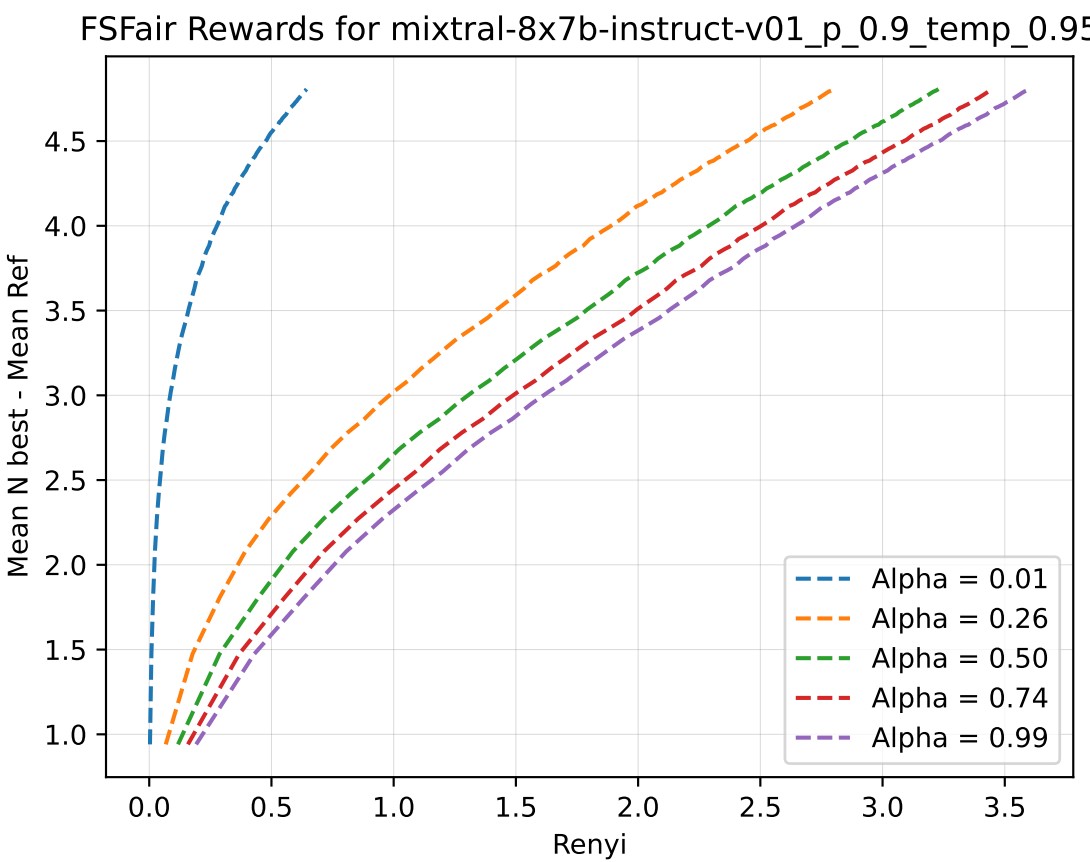

Figure 23: Centered Reward for Best of $N$ versus Renyi divergence for $\alpha \in (0, 1)$ -Mixtral

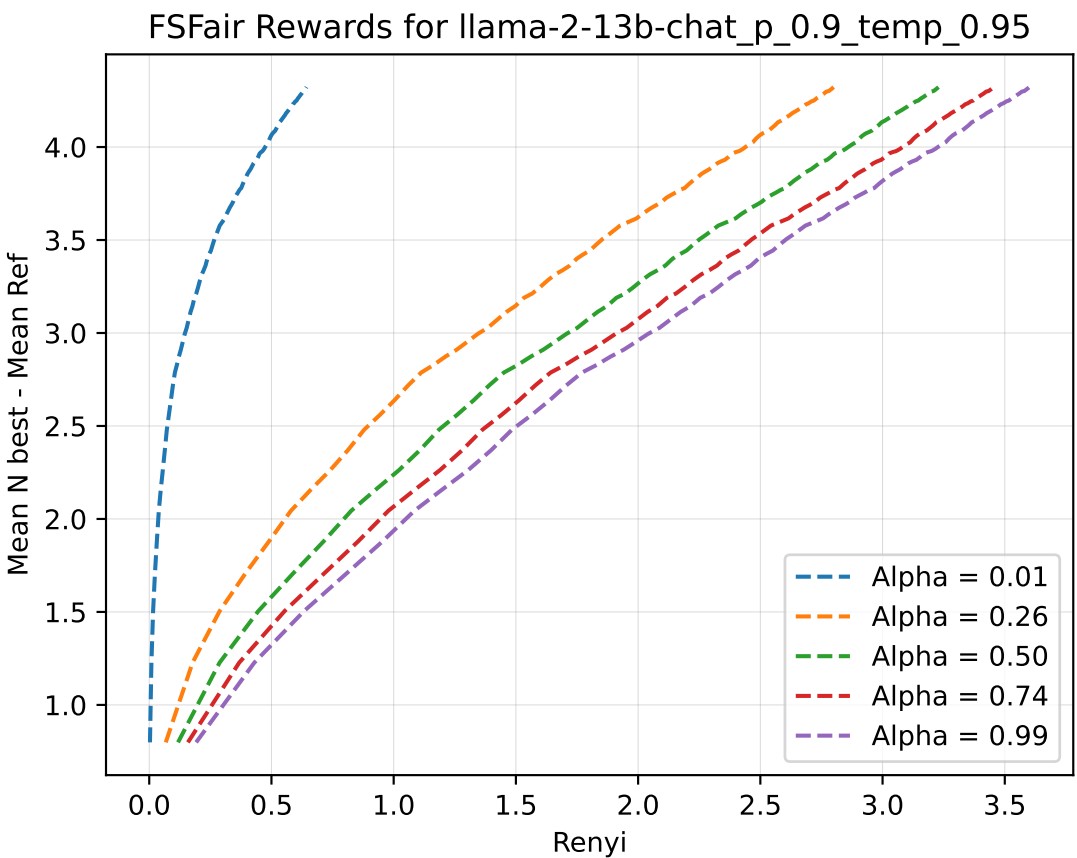

Figure 24: Centered Reward for Best of $N$ versus Renyi divergence for $\alpha \in (0,1)$ -LLama

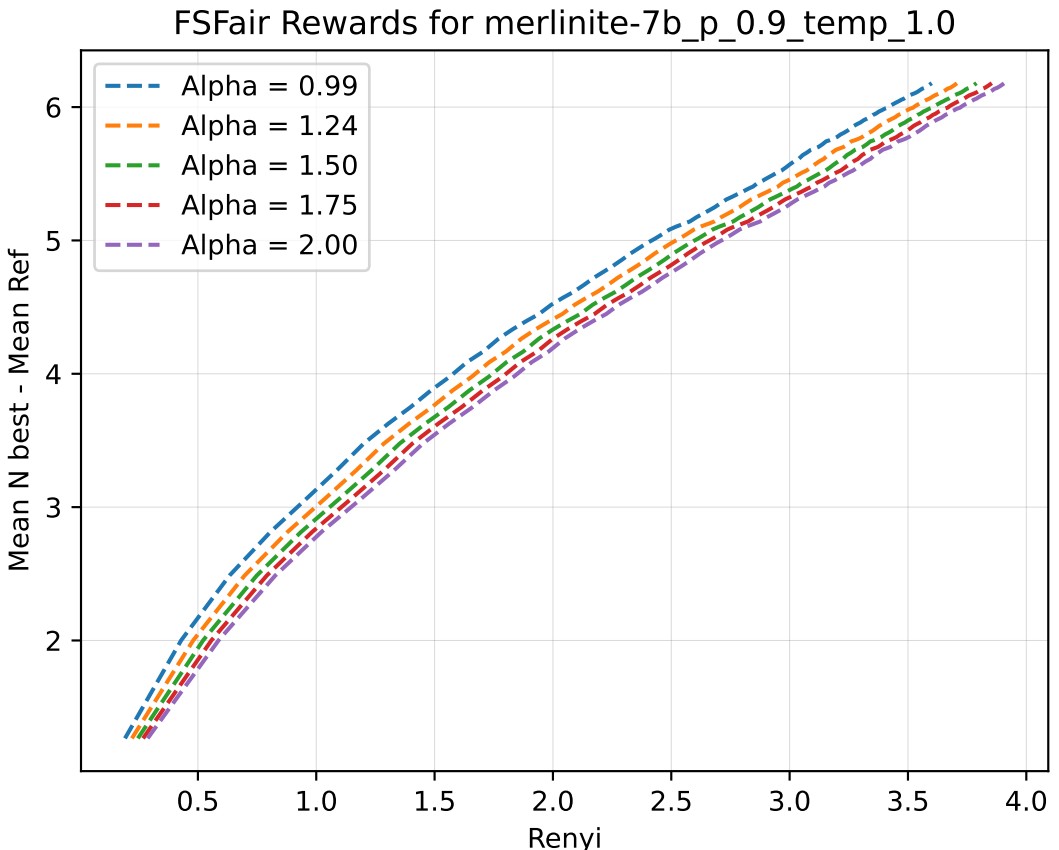

Figure 25: Centered Reward for Best of $N$ versus Renyi divergence for $\alpha > 1$ - Merlinite

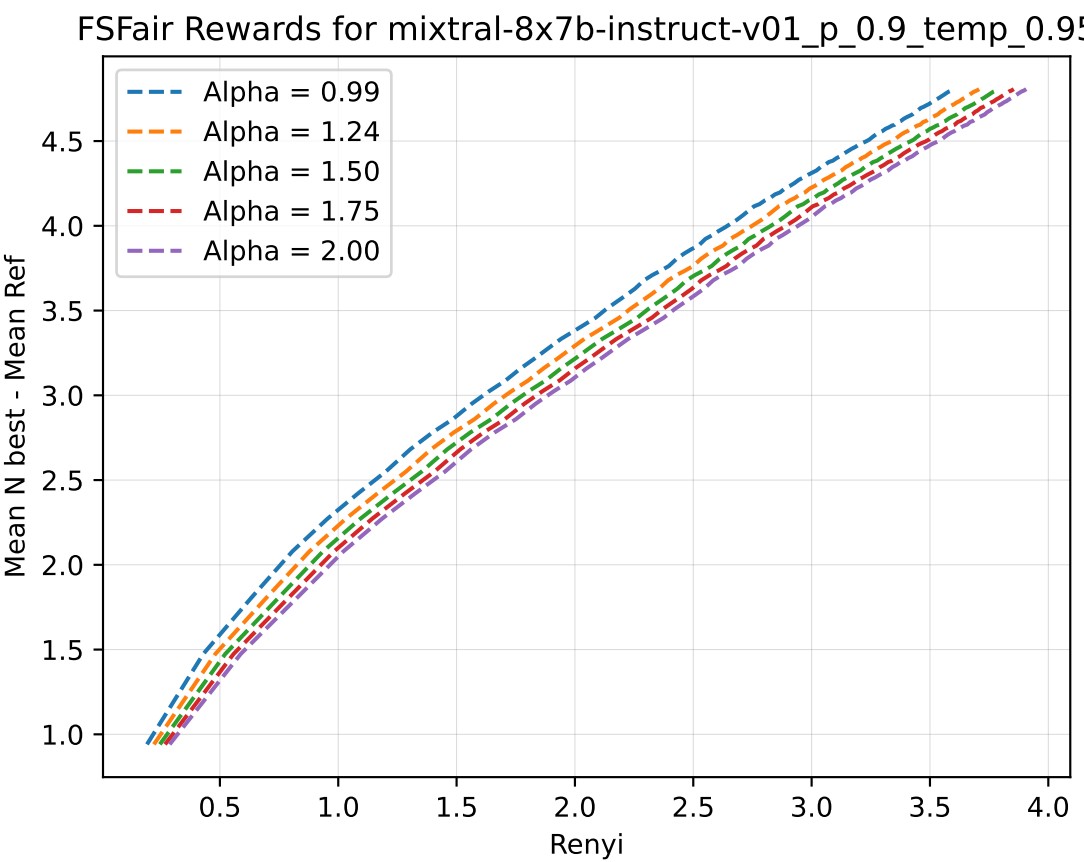

Figure 26: Centered Reward for Best of $N$ versus Renyi divergence for $\alpha > 1$ -Mixtral

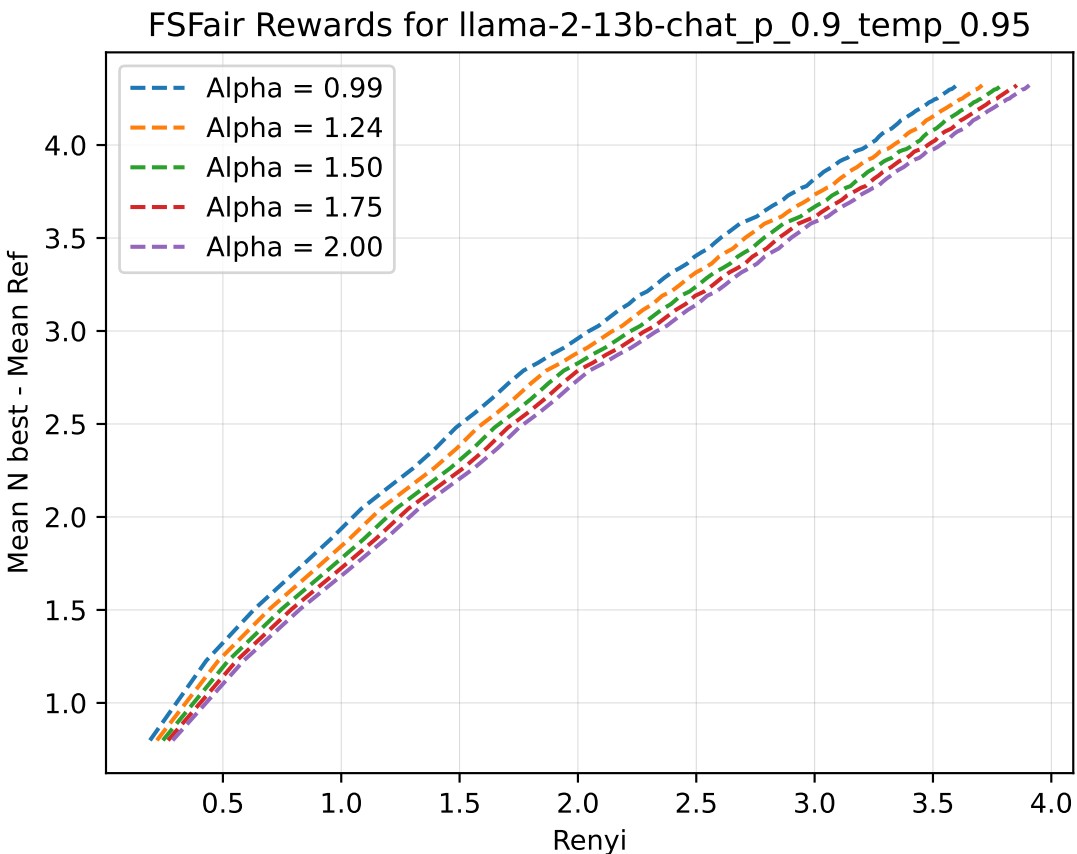

Figure 27: Centered Reward for Best of $N$ versus Renyi divergence for $\alpha > 1$ -LLama

