# OpenReview forum: "Information Theoretic Guarantees For Policy Alignment In Large Language Models"
_TMLR — Accepted by TMLR_

### Review · Reviewer_M4q8 · 2025-02-13

**Summary Of Contributions:**

This paper makes several theoretical contributions to Large Language Models (LLMs) alignment. The first contribution is to provide new proof for the best of $n$ policy inequality, extending previous work to more realistic settings and generalizing it to $f$-divergences and Rényi divergences. Second, it shows that the policy improvement scaling laws presented in Gao+ (2023) are information-theoretic upper bounds, derived from transportation inequalities with KL under sub-gaussian reward tails. Third, the authors examine the tightness of these bounds when the optimized policy’s tails are known, deriving new transportation inequalities for Rényi divergence. Finally, this paper studies the transfer of transportation inequalities from proxy to golden rewards, proving that golden reward improvement is limited by the proxy reward’s overestimation.

**Audience:**

Yes

**Broader Impact Concerns:**

This paper is purely theoretical, so I do not see any ethical concern in this paper.

**Claims And Evidence:**

Yes

**Requested Changes:**

I think this paper is sufficiently well-written in its current form, but I would like to request the following changes if possible:

- Rewrite Section 2.2 to be more reader-friendly. In my opinion, Beirami+ (2024) is a good reference for writing this section.
- Add empirical results for other divergences than reverse KL and see if we can observe similar consistency between theory and practice

**Strengths And Weaknesses:**

**Strength**
The authors did a great job in the field of LLM alignment. This paper is well-written and organized. While the math in this paper can sometimes be difficult to understand, given the difficulty of the content, it is easy to follow.

The main strength of this paper is its comprehensive information and theoretical analysis of policy alignment. The paper also provides a clear explanation of the technical material, making it accessible to a broad audience. I consider that some portion of this paper seems incremental compared to Gao+ (2023) or Beirami+ (2024), but I do feel that this paper poses solid contributions and it is an advantage to discuss the differences from such existing literature.

The theoretical analyses are interesting and will attract much attention from the research community. Best-of-$n$ test-time alignment is a promising paradigm, and this paper will contribute to understanding its influence and limitations theoretically.

Though the empirical evaluation is limited, it is very interesting to see that there is some consistency between theoretical and empirical results.

**Weakness**
I think this paper should be read by many researchers not only from ML theory but also from the empirical side of ML or NLP. The current form of this paper is partly hard to follow and could be written in a way friendly to a wider range of audiences. For example, Section 2.2 could be written to be reader-friendly. In my honest opinion, the quality of Section 2.2 is much worse than that of other sections. I even feel that this section is written in a hurry.

The empirical results are minimal. I feel that the empirical results are sufficient to support the authors' claims, it would be helpful to provide more results regarding different divergences presented in Table 1.

---

> ### Author Response · Authors · 2025-02-13
> **thank you !**
>
> We thank the reviewer for their positive comments on the paper and for their suggestions. We address in the following their comments:
>
> ___
>
> **Rewrite Section 2.2 to be more reader-friendly. In my opinion, Beirami+ (2024) is a good reference for writing this section.**
>
> Thank you for your suggestion we will improve the readability of this section. We wrote this Section in a way that highlights more the mathematical tools in use we will add more context as suggested.
>
> ___
> **Add empirical results for other divergences than reverse KL and see if we can observe similar consistency between theory and practice**
>
> We already provide those results in the paper in Section 5  under "Reward Versus Rényi divergence for Best of n", the figures are given in the appendix in  Figures 16, 17 and 18 and  in Figures  19, 20 and 21, and these are computed using Table 1.  For space restriction we had those in the appendix while mentioned in the main paper.

---

> ### Comment · Reviewer_M4q8 · 2025-03-03
> **Thank you for feedback**
>
> Thank you for providing authors' feedback.
>
> **Paper writing.** All other reviewers seem to have similar concerns about the writing of this paper. I'm looking forward to reading a revised version.
>
> **Empirical results.** I am sorry for missing the experimental results in the Appendix. Yes, the authors have already provided the results I mentioned in the initial review.
>
> I feel that the other reviewers make reasonable and constructive comments. I will wait for the authors to submit a revised version based on the reviewers' comments.

---

> ### Author Response · Authors · 2025-03-12
> **Thank you !**
>
> Thank you for the feedback, we appreciate your time and effort in the review ! We have uploaded a revision of the paper and rewrote Section 2.2 with more explanations and illustrations to make it more reader friendly !  (All edits in the paper are in red.)

---

> > ### Comment · Reviewer_M4q8 · 2025-03-14
> > **Thank you for revision!**
> >
> > I would like to appreciate the authors' careful revisions. I think the current manuscript is satisfactory and ready for publication.

---

### Review · Reviewer_JnBY · 2025-02-28

**Summary Of Contributions:**

This paper presents several information-theoretic guarantees on policy alignment.

- The first contribution of this paper is to establish an upper bound on the KL divergence (and other f-divergences) of best-of-n policy and reference policy under fairly general assumptions and relating these to the data processing inequality.

- Second, the authors characterize upper bounds on the improvement in expected reward as a function of the KL divergence (for sub-Gaussian distributions) and establish this to be an information-theoretic upper bound corroborating the existing empirical findings in the literature.

- They use their results to establish upper bounds on the reward improvement merely given the distribution of the reward.

Overall, the paper makes significant contributions to the theory of alignment and would be a nice addition to the literature.

**Audience:**

Yes

**Broader Impact Concerns:**

The paper studies the theory of alignment and I think broader implications have been sufficiently addressed by the authors

**Claims And Evidence:**

Yes

**Requested Changes:**

Please address the aforementioned weaknesses. A more detailed list of changes to address these issues is provided below.

- I think it would be very nice if the results are explicitly extended to the case where reward is calibrated in the sense of (Beirami et al., 2024, Eq. (28)). In this case reward is bounded on (0,1) and in fact is uniformly distributed over [0,1] so many of the results in this paper should simplify. In fact, these would imply guarantees on win rate (which is an important quantity of interest). Further, it would be nice to contrast the upper bounds on the calibrated reward with (Beirami et al., 2024, Conjecture 7.2) which basically says that n/(n+1) as a function of log(n) - (n-1)/n is an upper bound on the win rate.

- When discussing the unboundedness of the reward, in Section 3 end of Page 6, it would be good to mention that this may be an artifact of how we construct rewards in practice, and we may actually have to transform rewards to make them bounded. Besides, IPO (Azar et al., 2024) has already proposed to use win rate directly as reward which makes the reward bounded so a connection with IPO would also be nice. So I think the quote of Lambert et al. (2024b) might also be an overclaim in general.

- Page 2, end of Section 1: "Can we before doing any alignment predict the upper bound of any alignment from looking only on the histogram of the reward of the reference models?" -> "Without performing alignment, can we predict an upper bound on the expected reward for a given KL level given the distribution of the reward?"

- Please use \citet and \citep correctly. For example, when the sentence starts with a citation it should be \citet. Also, please consider using a different color than black for hyperlinks.

- Beginning of Section 2: \cal X and \cal Y are technically sets, not spaces.

- The constant before the KL regularizer is usually \beta in the literature whereas it is chosen to be 1/\beta in this paper. While this is fine given the paper is self-contained, I suggest to use \lambda or some other notation instead and in a footnote explain that 1/\lambda is usually called \beta in other papers.

- Please use \quad in Eq. (2), Eq. (3), etc to make the constraints more readable.

- End of page 3, not usually meet -> met

- Please expand on the context around (6); I think this is very hard to digest for a typical reader who doesn’t have the necessary background.

- Eq. (7) does not need to be broken in two lines.

- Please provide more context on DPI. I think majority of the readers of this paper are unfamiliar with (Polyanskiy & Wu 2023).

- I think the biggest value of Theorem 1 is that it applies to continuous distributions, e.g., diffusion models. In fact, the techniques of (Beirami et al., 2024) cannot be extended to handle this case. For example, (Singh et al., 2025) already build on Theorem 1 to analyze the KL divergence of best-of-n in diffusion models, which would not have been possible without Theorem 1 in this paper. Singh, Anuj, et al. "CoDe: Blockwise Control for Denoising Diffusion Models." arXiv preprint arXiv:2502.00968 (2025).

- After Theorem 1, it would be nice to connect the conditions for equality to the derivations of (Hilton & Gao), which basically implicitly assumes those conditions. Also, I think it would be good to mention that for LLMs with discrete outputs, those conditions cannot be satisfied so the inequality is always strict.

- The Eq. in Prop 4 does not need to be broken in two lines.

**Strengths And Weaknesses:**

The major strengths of the paper are the three aforementioned major contributions that are highly novel and non-trivial.

The major weakness of the paper are:
- The paper might be hard to follow for a non-expert in information theory. Given that this is a journal paper with no page limit, I think the authors can teach some of the basic concepts and their relevance before giving the results.

- Another weakness of the paper is that the majority of the results are derived for unbounded rewards that are sub-Gaussian. While this is a very reasonable assumption for the raw rewards, the actual golden reward of interest might indeed be bounded in [0, 1] and well-behaving, e.g., when the reward is calibrated through the CDF function such that the expected calibrated reward gives win rate (see Eq. (28) in (Beirami et al., 2024)).

---

> ### Author Response · Authors · 2025-03-12
> **Thank you !**
>
> Thank you for your detailed and insightful feedback! We really appreciate your time and effort in providing this feedback!  We have uploaded a revision of the paper in which we have addressed all questions.  **All edits are in red**. In particular:
> * All minor points  suggested references, typos , rephrasing have been addressed.
> * **We rewrote Section 2.2** making it more didactic and introducing illustrations and more context for optimal transport and data processing inequalities as suggested to make the paper more reader friendly as suggested
> * **for bounded and calibrated rewards** we added Section 3.3 that shows that $\sqrt{\mathsf{KL}}$ bounds are loose for this case and better bounds can be obtained via our $\mathsf{TV}$ upper bounds for best of $n$ and that better scalings with $\mathsf{KL}$ can be obtained further bounding total variation with the  Bretagnolle-Huber inequality.
> * We added in **Section 5 in page 15** experiments showing that indeed TV and  Bretagnolle-Huber KL upper bounds are the tightest for bounded and calibrated rewards.
> * We added a discussion after theorem 1, to explain the assumption done in  Hilton & Gao as suggested.
> * The constant in $\mathsf{KL}$ regularization : we changed it as you suggested and added a footnote to that effect.
>
> Thank you again and we are happy to discuss the edits in the paper !

---

> ### Comment · Reviewer_JnBY · 2025-03-14
> **Revisions are satisfactory**
>
> I'd like to thank the authors for the comprehensive revisions! After reading all reviews, I think this revision sufficiently addresses the major comments of the reviewers (including mine). I leave some minor comments that I think could be addressed in a subsequent minor revision of the paper.
>
> * In Eq. (3) and (4), please add blank space before "almost surely".
>
> * Please cite the theorem number for (6) from Polyanskiy and Wu.
>
> * Please cite the theorem that gives (9).
>
> * Please clarify that the techniques of (Beirami et al., 2024) could not be immediately extended to consider diffusion models. Perhaps both in contributions bullet points and when you mention (Singh et al., 2025). This is a major selling point of Theorem 1 in this paper.
>
> * Please also mention that the techniques of (Beirami et al., 2024) do not directly give all results in Table 1. Again, this is made possible due to the generality of the techniques used in this paper, which is another major selling point of the proof techniques in this paper.
>
> * Page 8, right before Sec 3.1, << -> \ll
>
> * At the beginning of Section 3.3, please expand how technically win rate and CDF transformations would look like. I think this is a bit too abstract for a reader who is not familiar with the cited works.
>
> * At the end of Section 3.3, it would be great to consider a toy continuous language model where the outcome is supported on [0, 1] and the reward is also ~U[0, 1]. In this case, the expected reward difference gives exactly win rate, which is exactly n/(n+1). It would be nice to compare the TV bound with this quanitity and perhaps plot all bounds as a function of KL which would be log(n) - (n-1)/n). I think generally, validating as many of results as possible on such toy example and perhaps others with a different distribution  on reward (e.g., Gaussian, exponential, etc) would be both simple and shed light on the results, tightness of the bounds, etc.

---

> > ### Author Response · Authors · 2025-03-14
> > **Thank you !**
> >
> > Thanks, we will implement those minor suggestions !  and thank you also for the last  great suggestion at end of section 3.3 we will add these shortly , they will be indeed very insightful!

---

### Review · Reviewer_y2Mw · 2025-03-01

**Summary Of Contributions:**

The paper studies information-theoretic guarantees of various LLM alignment methods, including training-time method KL-regularized reinforcement learning (KL-RL), and test-time method best-of-N. The main contributions are the following:

(1) Prove an upper bound on the KL divergence of the best-of-n policy $\pi^{(n)}$ against the base policy $\pi_{ref}$, which relaxes the assumption imposed in previous work.

(2) Prove an upper bound on the improvement of the reward for the aligned policy which depends on $\sigma_{ref}\sqrt{KL(\pi \|\|\pi_{ref})}$, where $\sigma_{ref}$ is the subgaussian proxy of the reward under the base model, improving the trivial bound which has the max norm of the reward instead of $\sigma_{ref}$. The paper also provide a tail-dependent upper bound on the improvement of the reward that depends on the Renyi divergence.

(3) The paper also studies the case where the learned reward is not the same as the golden reward, and provide an upper bound on the reward improvement taking the discrepancy between the two rewards into account.

**Audience:**

Yes

**Broader Impact Concerns:**

I don't see any particular ethical concerns of the work that needs additional discussion.

**Claims And Evidence:**

Yes

**Requested Changes:**

See previous comment on weakness.

Also there seems to be a many typos in the paper. So I would suggest the authors to do another round of proofreading.

List of typos I found:
1. The line after equation (1), missing parentheses in the subscript of the expectation symbol.
2. Section 2.2. First equation, after where, Y should be italicized.
3. Assumption 2: Right parentheses do not match left ones.
4. Beginning of page 4: conditionnaly -> conditionally or conditioned?
5. Lemma 1: $E \sim exp(1) $  -> $E \sim Exp(1) $.
6. After lemma 1: best n -> best of n.
7. The sentence is broken in proposition 1.
8. Page 6, notations and 'Scaling Laws in Alignment': $\sigma_{ref}$ is used without being defined.
9. Page 11, end of the hanging paragraph: missing a period.

**Strengths And Weaknesses:**

*Strengths*

The paper provides rigorous analysis of the theoretical guarantees on various alignment methods. Moreover, the theoretical results either provides theoretical justifications on some of the widely-used practical guidelines (e.g., the KL divergence between best-of-n policy and the base policy), or provides insights on some of the empirically observed phenomena in alignment (e.g. the square root dependence between the reward improvement and KL divergence).

The result are new to the best of my knowledge. I appreciate the rigorous approach that the authors take on studying these problems and I believe these results will be of interest to both the LLM alignment community and the information theory community.

I didn't verify all the proofs but the results look believable to me.

*Weaknesses*

The results on reward improvement are mainly upper bound results. It would be nice if the authors could also comment on the lower bound side to provide a more complete picture and justify the tightness of the result. For example, Proposition 4 shows that the upper bound on the reward improvement could be hurt by the discrepancy between the proxy reward and the real reward. While this hints on that the reward improvement might also be hurt, however, this does imply this fact. The argument holds similarly to other results such as proposition 3 as well.

The writing of the paper could be improved. This is mainly a stylistic suggestion. The paper contains a list of results with different techniques and assumptions. So there can be a few switching of topics through the paper. While I believe each one of the results is interesting, emphasizing on the a few and providing sufficient discussions on their implications might help the readers grasp the main message of the paper.

*Minor comments*

1. Can the authors comment on when Assumption won't hold?
2. The authors make the assumption that the optimization problem in KL-RL will be solved to its optimality, which may not be true in practice. This is mainly a comment on the limitation. I don't view this as a major weakness since many papers made the assumption.

---

> ### Author Response · Authors · 2025-03-12
> **Thank you !**
>
> Thank you for your positive feedback we appreciate your time and effort ! We have uploaded a revision of the paper , **all major edits are in red**, in particular:
>
> * We rewrote Section 2.2 to make it more reader friendly and with more illustration and explanations
> * We corrected all typos that you mentioned and proofread the paper.
> * **Regarding lower bounds and tightness of scaling laws**  we highlighted in page 9 a comment in the paper that says if the reward is subgaussian (upper bounded by a gaussian ) and lower bounded also by a gaussian then we have a matching lower bound.  Lower bounds for Gaussians were developped in  (Kamath, 2015)). We added also Remark 1 in page 9 that shows that  the KL scaling laws are tight for gaussians. For gaussians we have an equality instead of just an inequality.
> * **The assumption that the optimization problem in KL-RL will be solved to its optimality** For transportation inequality, we don't need the solution to be solved to optimality, since Corollary 1 is an application of proposition 3 that holds for any $\pi$, we added in Corollary one, item 2  (page 9) for a KL-RL solution that is suboptimal but feasible. Note that in this case we can not guarantee that this difference is positive and the guarantee is in absolute value.
> * **Can the authors comment on when Assumption won't hold?** We believe you are asking about assumption 2: this one is rather mild if we can have a probability kernel that gives $\mathbb{P}(Y|R,X )$, then assumption 2 is satisfied, since most reward models are modeled as $\mathbb{P}(R|Y,X)$ this can be  obtained using the bayes rule. We has this clarified this in page 4 under Assumption 2.

---

> > ### Comment · Reviewer_y2Mw · 2025-04-01
> >
> > Thanks the authors for the response and revisions. I think the current version is good and I would recommend for an accept.

---

### Author Response · Authors · 2025-03-03
**Thank you for your reviews**

Thanks to all the reviewers for their insightful and constructive comments, we will address all reviews point by point and submit within a week from now a revised version of the paper.

Best,

Authors

---

### Author Response · Authors · 2025-03-12
**Paper revision**

We thank all reviewers for their positive insightful and great feedback ! We really appreciate the time and effort you had put into this. We have uploaded a revision of this paper, all major edits are in red. In particular:
* We **rewrote Section 2.2** , to make the paper more reader friendly as suggested by reviewers. We introduced all tools needed from information theory and optimal transport with illustrations and with a better flow.
* We **addressed the case  of calibrated and bounded reward**  and showed in Section 3.3 that our $TV$ upper bounds provide the tightest scaling  laws for best of n reward improvements, and provided better scaling laws then the $\sqrt{KL}$ ones using Bretagnolle-Huber inequality. **We added in Section 5 experiments for calibrated rewards showing linear scaling laws with our TV upper bounds.**
* We discussed in **page 9 and in remark1 the tightness of the scaling laws**, and showed that lower bounds exist for best of n if the reward of the reference is lower bounded by a gaussian. We showed that these scaling laws are equalities for gaussians (when using one to one rewards and reward distributions are gaussians with same variance.)
* We clarified that transportation inequality for the RL problem don't need to have the RL problem solved to optimality, and clarified when our assumptions hold.
* We corrected typos and integrated all suggestions.

---

### Decision · Action_Editor_ZSBa · 2025-04-10

**Recommendation:** Accept as is

**Comment:**

This submission got clear positive reviews from all three reviewers, all recommending acceptance and stating that the theoretical contributions of this work will be of high interest to many ML and information theory researchers. The contributions are centered around a topic of high interest, which is understanding the expected improvements from policy alignment using existing popular techniques.

**Audience:**

Yes, this would be of interest to practitioners interested in understanding the empirical phenomena observed during post-training with RLHF and best-of-N and also to researchers from the information theory side interested in applications to LLMs.

**Claims And Evidence:**

All reviewers agree that this paper provides meaningful theoretical contributions that contribute to the understanding of post-training techniques such as RLHF and best-of-N. These results go beyond previous work, explain some commonly observed empirical findings and  are helpful for characterizing the improvement in expected reward under some assumptions on the reward disrtibution.

---

> ### Author Response · Authors · 2025-04-16
> **Thank you**
>
> Thanks to the editor and the reviewer for their time and their positive and insightful feedback ! We will start preparing the camera ready paper.